# Network segregation is associated with processing speed in the cognitively healthy oldest-old

Sara A Nolin[1]*, Mary E Faulkner[1], Paul Stewart[1], Leland L Fleming[1],
Stacy Merritt[2], Roxanne F Rezaei[3], Pradyumna K Bharadwaj[4],
Mary Kate Franchetti[4], David A Raichlen[5], Cortney J Jessup[4], Lloyd Edwards[1],
G Alex Hishaw[4], Emily J Van Etten[4], Theodore P Trouard[4], David Geldmacher[1],
Virginia G Wadley[1], Noam Alperin[2], Eric S Porges[3], Adam J Woods[3],
Ron A Cohen[3], Bonnie E Levin[2], Tatjana Rundek[2], Gene E Alexander[4],
Kristina M Visscher[1]*

[1]University of Alabama at Birmingham Heersink School of Medicine and Evelyn
F. McKnight Brain Institute, Birmingham, United States; [2]University of Miami
Miller School of Medicine and Evelyn F.McKnight Brain Institute, Miami, United
States; [3]University of Florida and Evelyn F. and William L.McKnight Brain Institute,
Gainesville, United States; [4]University of Arizona and Evelyn F. McKnightBrain
Institute, Tucson, United States; [5]University of Southern California, Los Angeles,
United States

*For correspondence:
nolin@musc.edu (SAN);
kmv@uab.edu (KMV)

Competing interest: The authors
declare that no competing
interests exist.

Reviewing Editor: Juan Helen
Zhou, National University of
Singapore, Singapore

**Abstract** The brain is organized into systems and networks of interacting components. The
functional connections among these components give insight into the brain's organization and may
underlie some cognitive effects of aging. Examining the relationship between individual differences
in brain organization and cognitive function in older adults who have reached oldest-old ages with
healthy cognition can help us understand how these networks support healthy cognitive aging. We
investigated functional network segregation in 146 cognitively healthy participants aged 85+ in the
McKnight Brain Aging Registry (MBAR). We found that the segregation of the association system
and the individual networks within the association system (the fronto-parietal network , cingulo-
opercular network, and default mode network), has strong associations with overall cognition and
processing speed. We also provide a healthy oldest-old (85+) cortical parcellation that can be used
in future work in this age group. This study shows that network segregation of the oldest-old brain
is closely linked to cognitive performance. This work adds to the growing body of knowledge about
differentiation in the aged brain by demonstrating that cognitive ability is associated with differenti-
ated functional networks in very old individuals representing successful cognitive aging.

## Editor's evaluation

This useful study provides solid support for how brain function at the system level, particularly
network segregation, influences cognitive abilities even in the oldest-old range of human aging. The
findings are potentially interesting to help understand successful aging.

## Introduction

It is an important societal goal to slow age-related cognitive decline. Understanding the factors that
contribute to optimal cognitive function throughout the aging process is essential to the development

of effective cognitive rehabilitation interventions. To better understand successful cognitive aging, we recruited participants who have reached oldest-old age (i.e., 85+ years old) with documented healthy cognition and examined the relationship between variability in behavior within this cohort and measures of their brain network segregation, large-scale patterns of functional connectivity measured with fMRI. Prior work has mostly been done in younger-old samples (largely 65–85 years old). Studies of the younger-old can be confounded by inclusion of pre-symptomatic disease since it is unknown which individuals may be experiencing undetectable, preclinical cognitive disorders and which will continue to be cognitively healthy for another decade. The cognitively unimpaired oldest-old have lived into late ages, and we can be more confident in their status as successful agers. Studying successful cognitive agers brings another advantage: given the aging process, as well as the years of experience they have due to their advanced age, there is greater variability in both their performance on neurocognitive tasks and their brain connectivity measures compared to younger cohorts (*Christensen et al., 1994*). This increased variance makes it easier to observe across-subject relationships of cognition and brain networks (*Gratton et al., 2022*). Prior work studying the healthy oldest-old indicates intact cognition in this age group is impacted by influences such as cognitive reserve (*Kawas et al., 2021*) and resistance to Alzheimer's disease-related neuropathology (*Biswas et al., 2023*; *Gefen et al., 2015*). We extend oldest-old aging research by increasing our understanding of the oldest-old brain and provide novel insight into the relationship between the segregation of networks and cognition by investigating this relationship in an oldest-old cohort of healthy individuals.

Some cognitive domains are particularly susceptible to decline with age, including processing speed, executive function, and memory (*Reuter-Lorenz, 2016*; *Spaan, 2015*). Processing speed refers to the speed with which cognitive processes, such as reasoning and memory, can be executed (*Sliwinski and Buschke, 1997*). *Salthouse, 1996* proposed that cognitive aging is associated with impairment in processing speed, which in turn may lead to a cascade of age-associated deficits in other cognitive abilities. Because processing speed is so strongly associated with a wide array of cognitive functions, it is crucial to understand how it can be maintained in an aging population. Executive functioning is a broad collection of cognitive capacities encompassing sustained attention, updating, inhibition, switching, and set-shifting (*Fisk and Sharp, 2004*; *Lamar et al., 2002*; *McCabe et al., 2010*; *Rabinovici et al., 2015*; *Sorel and Pennequin, 2008*). Executive functioning performance reliably declines in normal aging (*Fisk and Sharp, 2004*; *Harada et al., 2013*; *Reuter-Lorenz, 2016*; *Salthouse et al., 2003*; *Spaan, 2015*), and this decline is faster in older ages (*Zaninotto et al., 2018*). Memory is another well-studied cognitive domain that encompasses multiple processes, such as encoding, consolidation, and retrieval of information (*Huo et al., 2018*; *Zlotnik and Vansintjan, 2019*). Age-related decline in memory is reported subjectively by most older adults (*Craik, 2008*), with episodic memory being the most impacted by aging compared to other memory systems (*Luo and Craik, 2008*). The cognitive domains of working memory and language functioning are known to be vulnerable to the aging process as well. Working memory refers to the simultaneous temporary storage and active manipulation of information (*Stanley et al., 2015*). There is reliable evidence across studies that working memory gradually declines from early to late adulthood (*Kidder et al., 1997*; *Salthouse and Babcock, 1991*; *Stanley et al., 2015*; *Vaqué-Alcázar et al., 2020*). Language function, particularly language production, also undergoes age-related decline and is related to other cognitive functions affected by aging, including working memory and executive function (*Rizio and Diaz, 2016*).

Brain networks play a crucial role in aging, and older adults exhibit differences in brain structural and functional network integrity that impact network dynamics (*Marstaller et al., 2015*). Because of their correlation to cognitive performance, brain network dynamics have emerged as a major avenue to study aging and cognitive decline (*Andrews-Hanna et al., 2007*; *Antonenko and Flöel, 2014*; *Chan et al., 2014*; *Cohen and D'Esposito, 2016*; *Ng et al., 2016*; *Shine et al., 2016*; *Wen et al., 2011*). Many properties of networks can be quantified to describe their overall structure, connectedness, and interactions with other networks (*Bullmore and Sporns, 2009*; *Damoiseaux, 2017*; *Thomson, 1939*). Within-network integration describes how much the network's regions interact and can be quantified as the mean connectivity of nodes within a given network (within-network connectivity). The network participation coefficient describes the amount of variety of connections of a given node. A low participation coefficient indicates a node is more selectively connected to its network, and high participation coefficient indicates a node is widely connected to other networks (*Rubinov*

*and Sporns, 2010*). Modularity describes how separable a system is into parts (*Rubinov and Sporns, 2010*). Lastly, segregation describes the balance of within and between network connectivity. Very high segregation indicates isolated networks, and very low segregation indicates the networks are no longer separable (*Wig, 2017*).

A neural system's functional segregation is determined by the network's balance of connections between and within the network and is indicative of organizational integrity (*Chan et al., 2017*; *Damoiseaux, 2017*; *Iordan et al., 2017*; *Koen et al., 2020*; *Varangis et al., 2019*). In older adults, functional networks have increased between-network connectivity and decreased within-network connectivity, which in turn decreases segregation (*Chan et al., 2017*; *Damoiseaux, 2017*; *Iordan et al., 2017*; *Koen et al., 2020*; *Varangis et al., 2019*). Prior research suggests various hypotheses about age-related cognitive decline, with one prominent theory being the dedifferentiation hypothesis. This hypothesis posits that as we age, brain networks lose their specialized functions, becoming less selectively organized and more homogenous in their activity (*McDonough et al., 2022*). In younger individuals, distinct brain regions tend to engage in specific tasks with maintained functional boundaries. However, with aging, these boundaries may blur, leading to a reduction in functional segregation—wherein brain networks that were once highly separable begin to overlap and interact more frequently in a less efficient manner (*Chan et al., 2014*; *Daselaar et al., 2015*; *Seider et al., 2021*; *Siman-Tov et al., 2016*). This reduced segregation is thought to contribute to cognitive decline by diminishing the brain's ability to process information in a targeted and efficient way. In our analyses, we examine segregation alongside other network organization metrics to understand how these network metrics manifest in the oldest-old. Our aim is not only to explore whether general network organization metrics are linked to cognition in this age group but also to investigate the potential evidence for the dedifferentiation hypothesis, contributing to our understanding of the neural processes involved in cognitive aging.

The health of brain networks, in particular their ability to have independent, or differentiated activity is thought to contribute to cognitive performance. Previous studies have found that dedifferentiation of higher-order cognitive networks of the association system —the fronto-parietal network (FPN), cingulo-opercular network (CON), and default mode network (DMN)—are related to poorer performance in many cognitive abilities, including episodic memory, processing speed, attention, and executive function (*Chan et al., 2017*; *Damoiseaux, 2017*; *Goh, 2011*; *Hausman et al., 2020*; *Iordan et al., 2017*; *Koen et al., 2020*; *Nashiro et al., 2017*; *Ng et al., 2016*; *Varangis et al., 2019*). The FPN is associated with complex attention and directing cognitive control (*Avelar-Pereira et al., 2017*; *Malagurski et al., 2020*; *Oschmann and Gawryluk, 2020*; *Ray et al., 2019*). The CON is associated with sustained executive control and perceptual and attentional task maintenance (*Coste and Kleinschmidt, 2016*; *Hausman et al., 2020*; *Sadaghiani and D'Esposito, 2015*). The DMN is activated during rest, internally focused tasks, and memory processing, but is suppressed during cognitively demanding, externally focused tasks (*Avelar-Pereira et al., 2017*; *Hampson et al., 2006*; *Hellyer et al., 2014*; *Ng et al., 2016*; *Sambataro et al., 2010*; *Sestieri et al., 2011*). Processing speed has been shown to be related to all of these networks (*Ruiz-Rizzo et al., 2019*; *Sheffield et al., 2015*; *Staffaroni et al., 2018*; *Vatansever et al., 2017*).

The purpose of this study was to understand the underlying brain network relationships associated with preserved cognition in oldest-old adulthood. Our examination of this cohort of individuals in the oldest age group addresses the gaps in previous research on aging. Firstly, existing studies often omit a growing segment of the elderly population by concentrating on brain networks in individuals under the age of 85. To enhance our understanding of the relationship between cognition and brain networks in the context of healthy aging, we have expanded previous network dynamics methods to encompass the oldest-old age range. Secondly, our study focuses on a sample of healthy oldest-old individuals, ensuring confidence in their status as successful agers due to their cognitive well-being at an advanced age. Thirdly, the greater variability in cognitive and brain network variables within our sample facilitates the observation of relationships across subjects, as demonstrated in previous studies (*Christensen et al., 1994*; *Gratton et al., 2022*). Lastly, the healthy oldest-old individuals epitomize the pinnacle of cognitive aging, having attained the expected lifespan without typical cognitive decline or the onset of cognitive disorders. This research provides valuable insights into the brain functioning of these relatively uncommon individuals, contributing to our understanding of preserved cognition into late life.

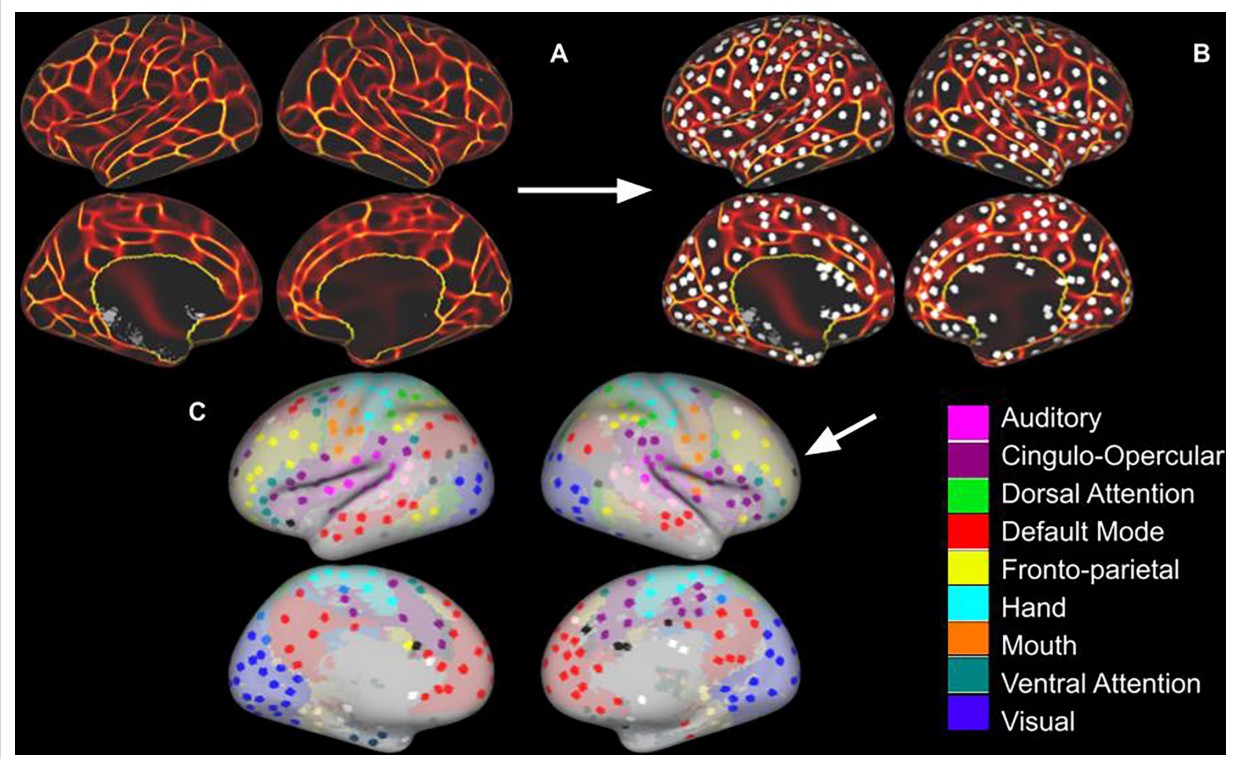

**Figure 1.** Regions of Interest Identification. (**A**) Functional connectivity boundary maps based on methods used by *Han et al., 2018*. (**B**) Local minima ROIs (Regions of Interest, 3 mm discs) based on methods used by *Chan et al., 2014*. (**C**) Local minima ROIs with the color of network membership of ROIs based on parcellation colors that are shown underneath ROIs (*Power et al., 2011*). White ROIs indicate nodes that do not belong to any labeled network.

The online version of this article includes the following figure supplement(s) for figure 1:

**Figure supplement 1.** Relationship between age and cortical thickness.

Here, we addressed the hypothesis that maintaining higher levels of cognitive function into healthy aging relies on greater segregation of the association system and its subnetworks: FPN, CON, and DMN. We predicted that segregation would be related to cognition and that other network organization metrics will have relatively weaker associations. We predicted that lower segregation within the Association System, FPN, CON, and DMN would be related to poorer overall cognition and cognitive domain performance in oldest-old adults. We used partial correlations between cognitive measures and network properties to test their association in this oldest-old aged cohort.

## Results

### A priori power analysis

A power analysis was performed which indicated that with a sample size of 146, an alpha of 0.05, and a power of 0.80, all analyses can detect small effect sizes, with the smallest detectable effect for a correlation being $r = 0.23$. This result indicated this study is sufficiently powered to detect results similar to the effect size found by *Chan et al., 2014*.

### Exploratory factor analysis

Exploratory factor analysis (section 'Cognitive measures') revealed five cognitive factors: (1) processing speed, (2) episodic memory, (3) executive functioning, (4) working memory, and (5) language (see *Supplementary file 1* for variable factor loadings). Overall cognition was calculated as the average of an individual's factor scores across the five factors.

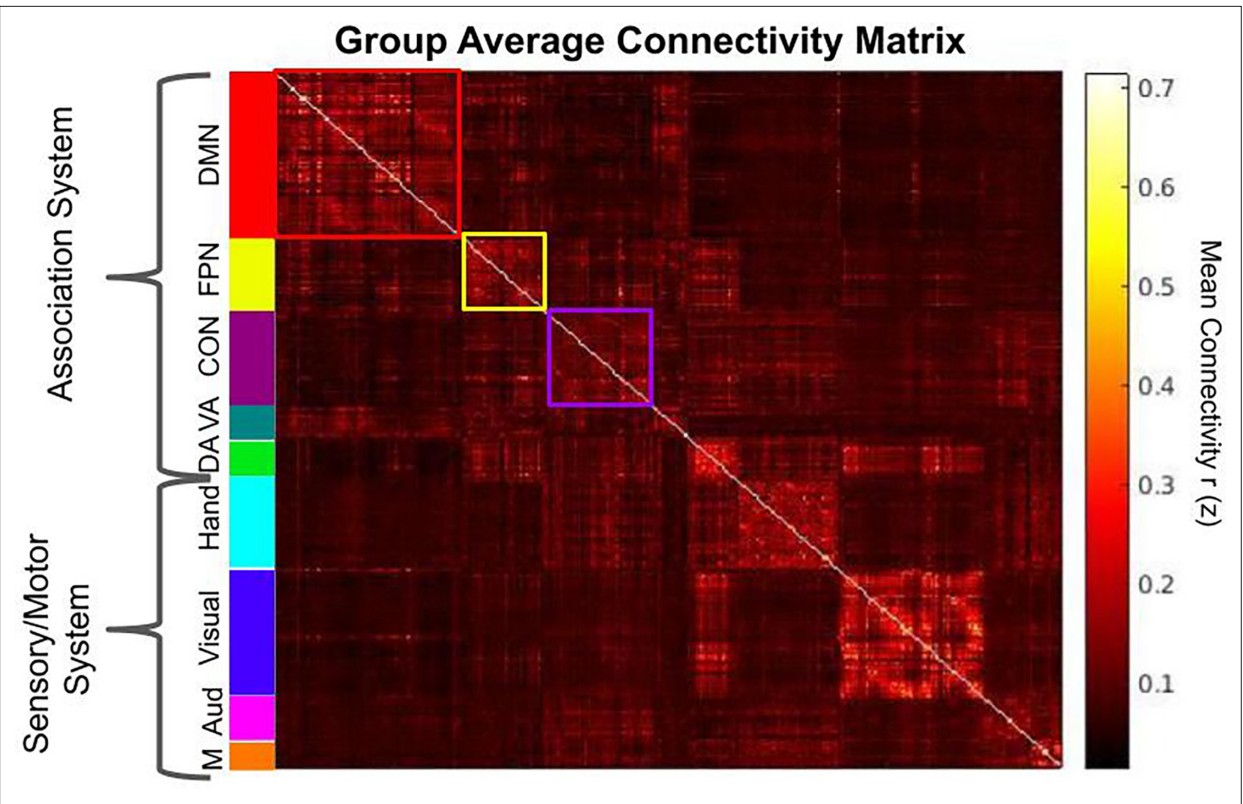

**Figure 2.** Group average Fisher's z- transformed correlation matrix of 321 nodes. The association system consists of the default mode (DMN; red), fronto-parietal control (FPN; yellow), ventral attention (VA; teal), cingulo-opercular control (CON; purple), and dorsal attention (DA; green). The sensory-motor system consists of the hand somato-motor (light blue), visual (blue), mouth somato-motor (M; orange), and auditory networks (Aud; pink).

## Functional connectivity of network nodes

We created network nodes based on methods developed by *Chan et al., 2014* and *Han et al., 2018* for our oldest-old sample (*Figure 1*, section 'Network nodes'). For comparison analysis, we also used the nodes created by *Chan et al., 2014* and *Han et al., 2018*. As an additional comparison parcellation, we used nodes that were created with MBAR data (*Figure 1*), then used Louvain algorithm-based community detection to assign node membership.

Using the ROIs we created (*Figure 1*), we generated a group average of Fisher's z-transformed correlation matrix grouped by network and system membership (*Figure 2*, section 'Calculation of network properties').

## Association system metrics and overall cognition

We then generated descriptive statistics of association system metrics and the overall cognition metric (*Table 1*; section 'Network analysis'). Our sample had a mean association system segregation of 0.4205, which is consistent with previous older adult cohorts from *Chan et al., 2014* (n =

**Table 1.** Descriptive statistics for association system metrics and overall cognition metrics.
All variables are unitless except mean within-network connectivity (z-score).

| Association system and overall cognition metrics | Mean | SD | Range |
|---|---|---|---|
| Segregation | 0.4205 | 0.1071 | 0.0929–0.6463 |
| Mean within-network connectivity (z-score) | 0.0833 | 0.0246 | 0.0162–0.1522 |
| Participation coefficient | 0.4356 | 0.0235 | 0.3675–0.4746 |
| Modularity | 0.2561 | 0.0374 | 0.1321–0.3501 |
| Overall cognition factor score | 0.00989 | 0.4428 | –0.96–1.4 |

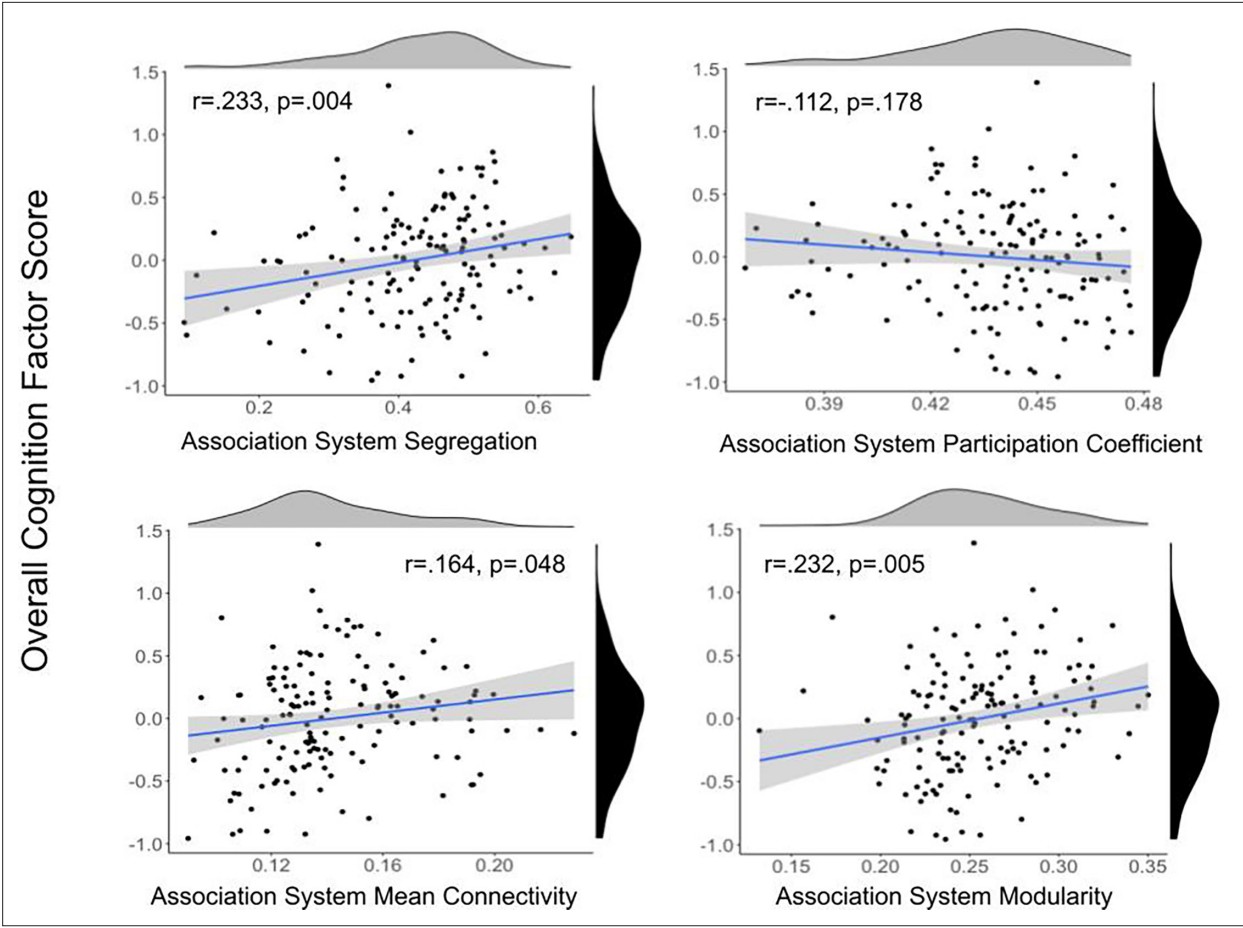

**Figure 3.** Scatter plots between association system metrics and overall cognitive performance. Density plots for the variables are presented for each variable on the edge of the scatter plot. Overall cognition score is shown in black, and association system metrics are shown in gray. Overall cognition was related to association system segregation, modularity, and mean within-network connectivity, but not participation coefficient. Only the relationship between overall cognition and segregation and modularity remained significant after multiple comparisons correction using false discovery rate (FDR).

60, 65–89 years) mean association system segregation of 0.40 (***Chan et al., 2014***; ***Figure 6—figure supplement 1***), and ***Han et al., 2018*** (n = 46, 80–93 years) mean association system segregation of <0.50 (***Han et al., 2018***; Figure 7). Additionally, these data fit with the general trend of decreasing segregation with age (***Figure 6—figure supplement 1***). These values are more reliable when the nodes are age-appropriate (***Han et al., 2018***), thus to control for possible differences due to the differences in the nodes, we also report the segregation values calculated using the same nodes used in Chan et al. (***Figure 6—figure supplement 1***). The pattern of results is similar regardless of which node set is used, though the age-appropriate nodes result in stronger segregation values.

In order to address the prediction that segregation will be related to cognition and that other network organization metrics will have relatively weaker associations, we analyzed the relation between graph theoretical metrics and cognition. Overall cognition was related to association system segregation ($r$ = 0.233, p=0.004), modularity ($r$ = 0.232, p=0.005), and mean within-network connectivity ($r$ = 0.164, p=0.048), but not participation coefficient ($r$ = –0.112, p=0.178) (***Figure 3***, section 'Relating cognition to network metrics'). Only segregation and modularity remained significant after multiple comparisons correction using false discovery rate (FDR) (***Benjamini and Hochberg, 1995***). Results of partial correlations with site and cortical thickness of the association system nodes as covariates indicated correlations were still significant and the effect size of the correlations remained largely unchanged with partial correlations. There was a strong, significant relationship between association system segregation and modularity ($r$ = 0.573, p<0.001). Multiple linear regression with these variables was also performed, indicating association system metrics were significantly associated with overall cognition (***Supplementary file 5***).

**Table 2.** Descriptive statistics of network segregation for each network.
All variables are unitless.

| Network segregation | Mean | SD | Range |
| --- | --- | --- | --- |
| DMN | 0.4517 | 0.1495 | 0.0628–0.7209 |
| FPN | 0.3279 | 0.1385 | −0.0482–0.5989 |
| CON | 0.3371 | 0.1330 | −0.0385–0.7062 |

CON, cingulo-opercular network; DMN, default mode network; FPN, fronto-parietal network.

## Network metrics and overall cognition

We then investigated the relationship of overall cognition with the network segregation of three networks that belong to the association system: FPN, CON, and DMN (*Table 2* and *Figure 4*). Of note, all significant network segregation relationships remain significant after correction for multiple comparisons using FDR. Partial correlation showed that the addition of cortical thickness and site as covariates did not impact the relationship between overall cognition and network segregation.

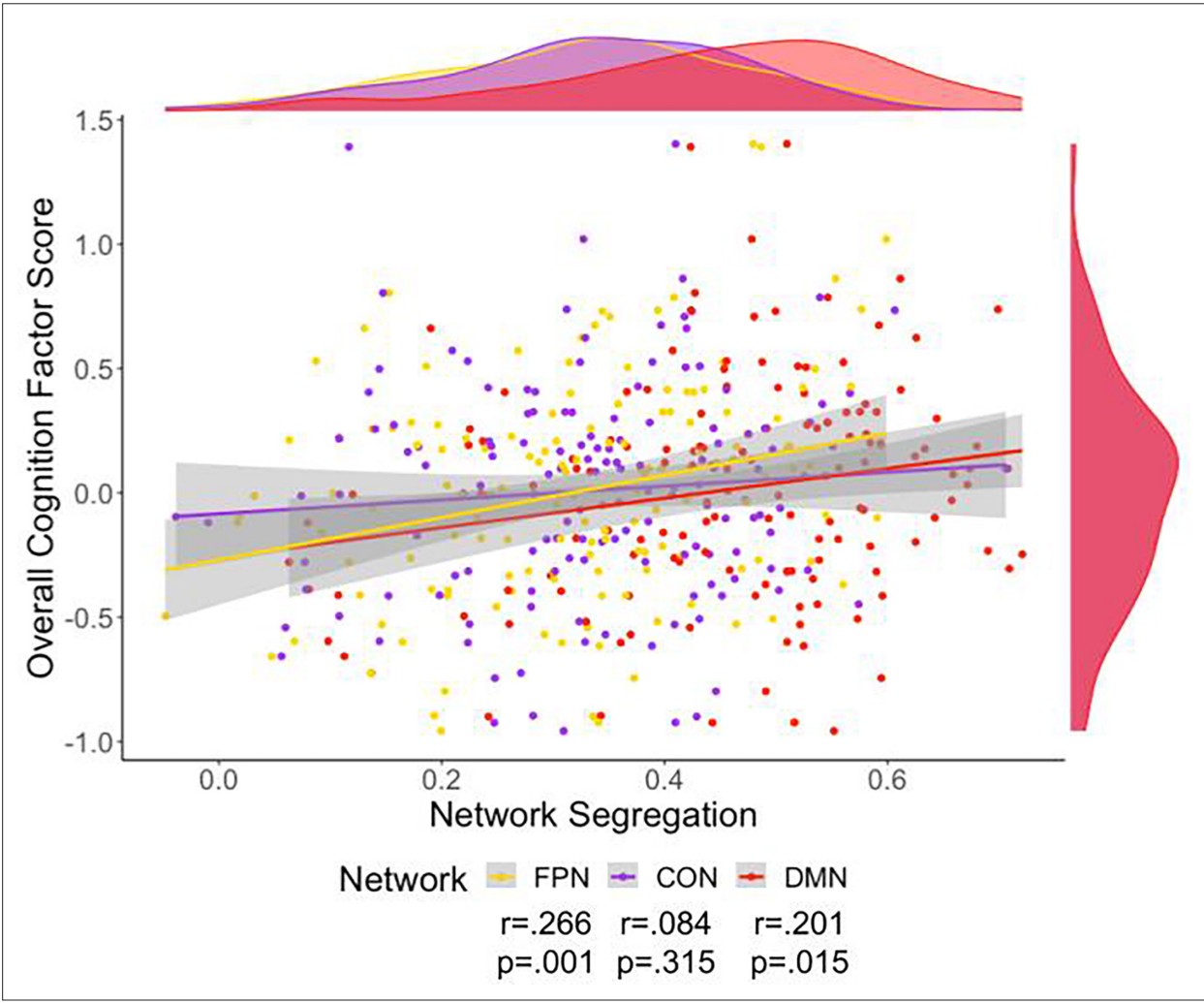

**Figure 4.** Scatter plot of overall cognition and fronto-parietal network (FPN) (yellow), cingulo-opercular network (CON) (purple), and default mode network (DMN) (red) network segregation. Density plots for the variables are presented for each variable on the edge of the scatter plot. The colors on these plots match the network color in *Figure 1*. Only cognition's relationship to segregation for FPN and DMN were still significant after adding a covariate of cortical thickness.

**Table 3.** Descriptive statistics of cognitive domain factor score as computed through the exploratory factor analysis.
All variables are unitless.

| Cognitive domain factor scores | Mean | SD | Range |
| --- | --- | --- | --- |
| Processing speed | 0.05 | 0.86 | –2.49–2.74 |
| Executive functioning | 0.05 | 0.83 | –2.36–2.34 |
| Episodic memory | 0.05 | 0.84 | –1.82–1.69 |
| Working memory | –0.02 | 0.87 | –2.4–2.53 |
| Language | –0.08 | 0.79 | –1.77–2.54 |

Multiple linear regression with these variables was also performed, indicating network segregation was significantly associated with overall cognition (*Supplementary file 5*).

### Network metrics and cognitive domains

In order to further address our prediction regarding the relationship between cognitive domains and network segregation in which lower segregation is related to poorer cognitive performance, we

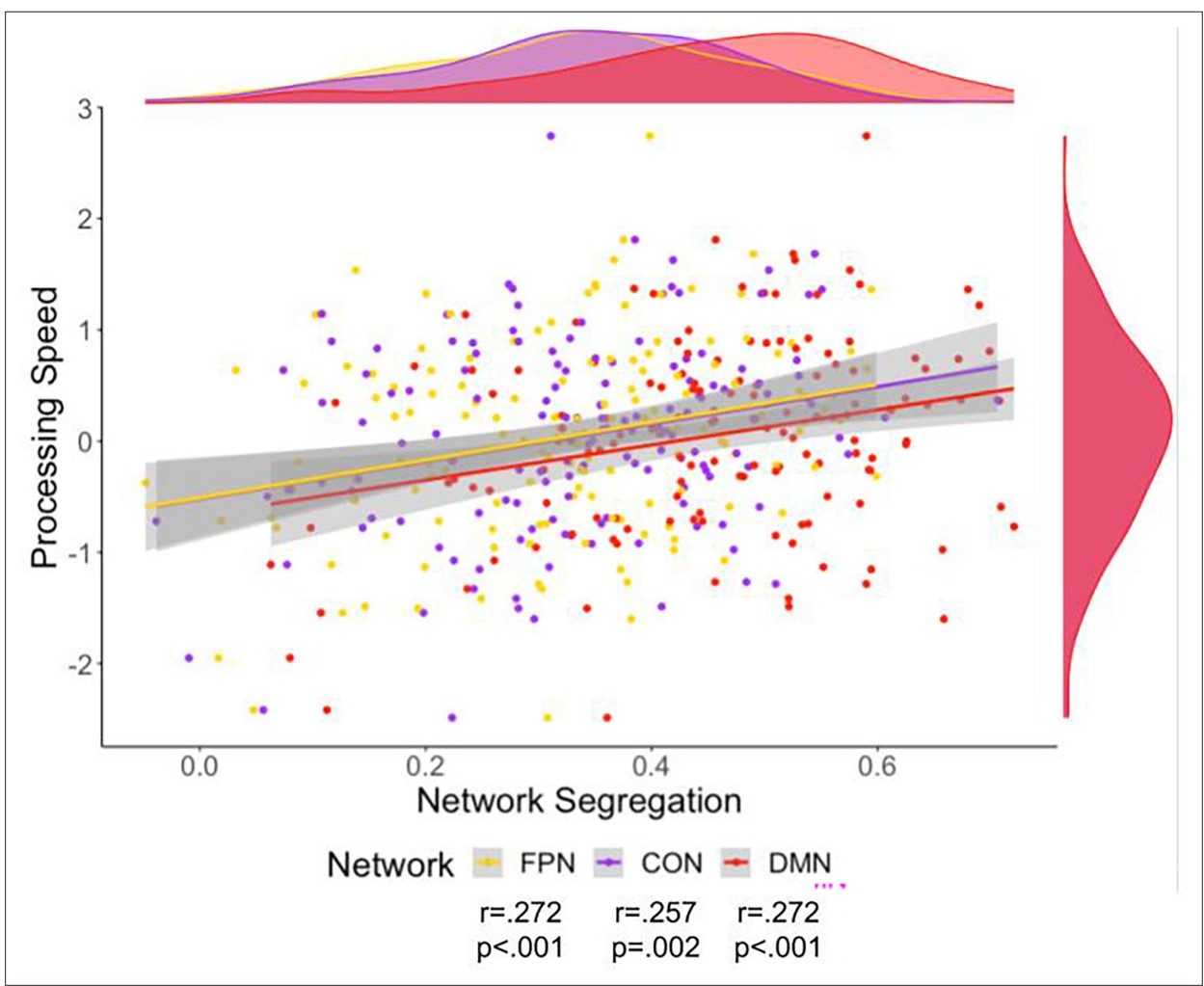

**Figure 5.** Scatter plot of processing speed and fronto-parietal network (FPN) (yellow), cingulo-opercular network (CON) (purple), and default mode network (DMN) (red) network segregation. Density plots for the variables are presented for each variable on the edge of the scatter plot. The colors on these plots match the network color in *Figure 1*. There was a significant relationship to Processing Speed for segregation of each of the networks.

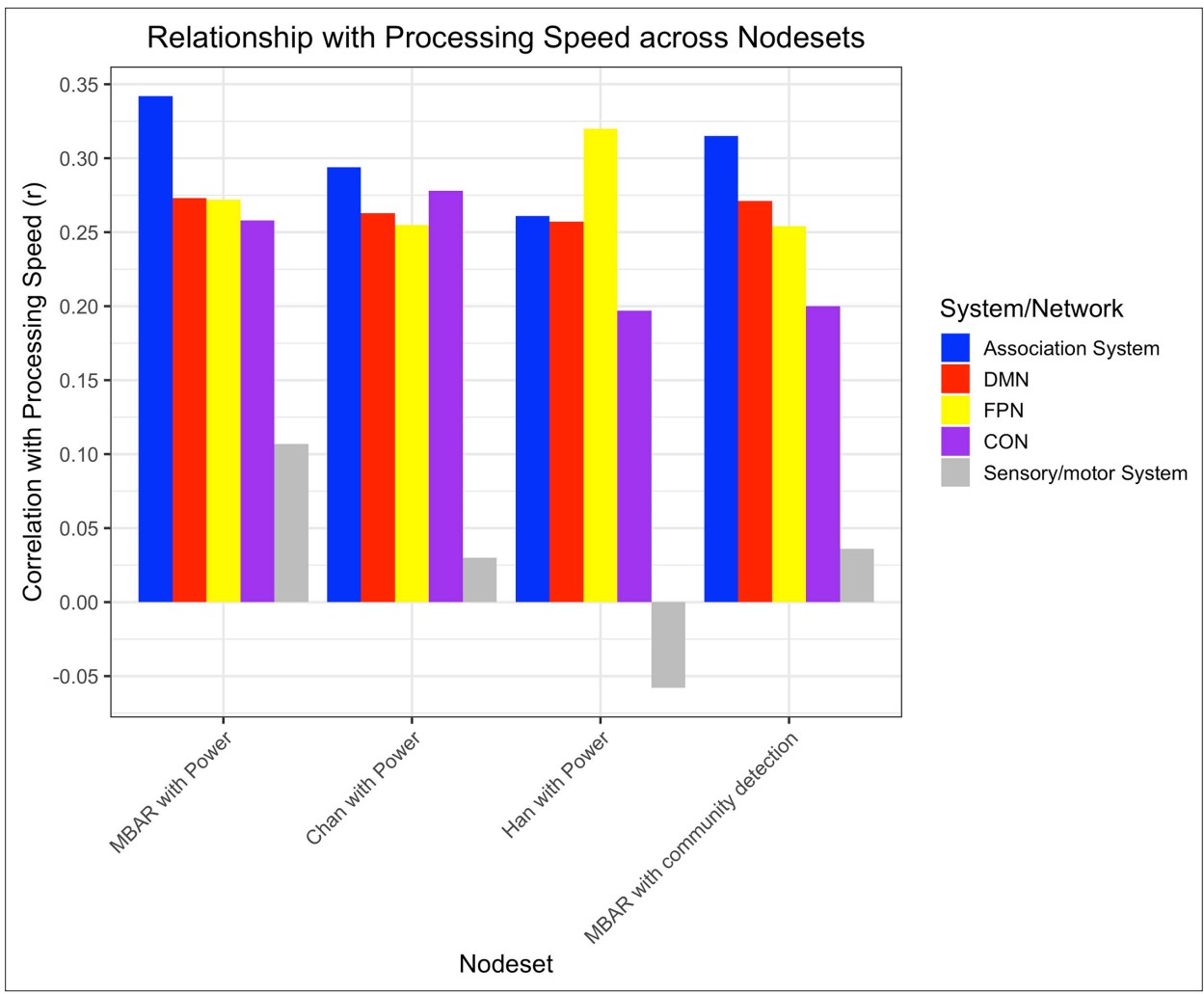

**Figure 6.** Bar plot of the correlation between processing speed and segregation within the association system, default mode network (DMN), fronto-parietal network (FPN), and cingulo-opercular network (CON), and sensory/motor system for each parcellation set. 'MBAR with Power' indicates node set created with MBAR data used to define the nodes, and *Power et al., 2011* (*Chan et al., 2014*; *Han et al., 2018*) atlas was used to determine node network membership (*Figure 1*). 'Chan with Power' indicates younger adults data used to define nodes (*Chan et al., 2014*; *Han et al., 2018*), and *Power et al., 2011* atlas used to determine node network membership. 'Han with Power' indicates older adults data from a different study (*Chan et al., 2014*; *Han et al., 2018*) used to define nodes, and *Power et al., 2011* atlas used to determine node network membership. 'MBAR with community detection' indicates MBAR data used to define nodes and MBAR data-based community detection used to determine node network membership (*Figure 7*). Sensory/motor system is included as a negative control and was not significant in any parcellation; all other correlations were significant.

The online version of this article includes the following figure supplement(s) for figure 6:

**Figure supplement 1.** Mean association system segregation across age.

investigated the relationship between the segregation of the FPN, CON, and DMN and five domains of cognition: processing speed, executive functioning, episodic memory, working memory, and language (*Table 3*). To compare with prior findings, we analyzed the relationship between association system segregation and memory, which was not correlated as had been previously found in the work by *Chan et al., 2017* ($r = -0.02$, p=0.805).

Processing speed was related to all networks' segregation (*Figure 5*). These relationships were still significant after correction for multiple comparisons using FDR, partial correlation with site as a covariate, and partial correlation with cortical thickness as a covariate. Additionally, these correlations were evident when using the alternative parcellations indicating replicability of these results (*Figure 6* and *Supplementary file 3*). Multiple linear regression with these variables was also performed, indicating network segregation was significantly associated with processing speed (*Supplementary file 5*).

Executive functioning was only significantly related to FPN segregation. However, this relationship was no longer significant after multiple comparison corrections. Correlations between executive functioning, memory, working memory, and language and the FPN, DMN, and CON segregation were very weak and not significant (*Supplementary file 4*).

## Discussion

Our study of this oldest-old sample fills in gaps of prior aging research. (1) Prior studies have excluded an ever-growing portion of the older adult population when studying network dynamics (these studies typically focus on people under age 85). We have extended prior methods in network dynamics to the oldest-old age range to better understand how aspects of cognition are related to brain networks in the context of healthy aging. (2) In studies of young-older adults (65–80), undetectable pre-symptomatic disease can confound results. In our sample of the healthy oldest-old, we can be confident in their status as successful agers since they are cognitively unimpaired at a late age. (3) More variability in cognitive and brain network variables makes it easier to observe across-subject relationships (*Christensen et al., 1994*; *Gratton et al., 2022*). (4) The healthy oldest-old represent the acme of cognitive aging since they have managed to reach expected lifespan without typical levels of diminished cognitive health or developed cognitive disorders. A limited number of studies have examined the healthy oldest-old (*Biswas et al., 2023*; *Gefen et al., 2015*; *Kawas et al., 2021*). These have shown factors which might confer resilience to age-related declines. However, these have not focused on resilience of functional networks, which are known to relate strongly to cognition. Since the function of these networks may mediate the relationship between anatomic pathology and cognitive function, examination of these networks is an essential step. This work gives us insight into the brain functioning of these relatively rare individuals and helps guide our understanding of how cognition is preserved into late ages.

First, we created a set of parcels for oldest-old adults based on functional connectivity boundary-based mapping. We then showed that association system segregation and modularity were related to overall cognition. We found that FPN and DMN network segregations were related to overall cognition, and FPN, CON, and DMN network segregations were related to processing speed. These results demonstrate that the oldest-old brain is segregated within the association system and the cognitive networks are important in supporting cognitive function and processing speed in the aged brain.

### Healthy oldest-old network parcellation

It is important to understand how a healthy aging cortex is subdivided, especially since brain network organization can change across the lifespan (*Bagarinao et al., 2019*). Previous work has measured brain organization in younger age ranges by creating boundaries between brain regions using shifts in functional connectivity patterns, boundary-based mapping, and then identifying nodes within those boundaries (*Chan et al., 2014*; *Han et al., 2018*). With the sample from the MBAR, we had the opportunity to apply the same methods to a sample with an older age range and larger sample size than previous work for the oldest-old portion of the sample. We provide a healthy oldest-old (85+) parcellation that can be used in future work in this age group and can be used to compare to disease populations in this age range. An age-appropriate parcellation may more accurately identify cortical mapping of networks. Future work will analyze the organization of the nodes in this parcellation and identify networks without younger-adult-based network descriptors.

### Age-related functional dedifferentiation

An influential theory for cognitive aging is the age-related dedifferentiation model which posits that functional networks in aging are not as selectively connected, or selectively recruited during tasks in older adults (*Goh, 2011*; *Koen et al., 2020*; *Li et al., 2001*; *Rakesh et al., 2020*; *Reuter-Lorenz and Cappell, 2008*; *Reuter-Lorenz et al., 1999*). We can quantify the level of differentiation by measuring functional segregation in brain network activity (*Chan et al., 2014*; *Daselaar et al., 2015*; *Seider et al., 2021*; *Siman-Tov et al., 2016*). Using the segregation metric, we can inform the dedifferentiation hypothesis.

The study of the association system and association networks across the lifespan has indicated that dedifferentiation is related to age and a co-occurring decrease in cognitive functioning (*Chan et al.,*

*2014*; *Geerligs et al., 2015*; *Han et al., 2018*). Longitudinal work on association system networks has indicated that segregation of association system networks decreases with age (*Chong et al., 2019*), and this rate of decline corresponds to declining cognitive functioning in the elderly (*Malagurski et al., 2020*; *Ng et al., 2016*). However, the mean age of participants in prior work was well below that of the current study, and the study sample size for the oldest-old was smaller than that of the current study. Therefore, it was unknown how far in the aging process dedifferentiation can continue while cognitive functions are maintained and to what degree different networks are sensitive to dedifferentiation in the oldest-old brain.

The goal of this study was to further investigate cognition and brain network differentiation in the context of successful brain aging in the oldest-old cohort by examining the metrics of segregation, participation coefficient, modularity, and within-network connectivity of the association system as well as the segregation of individual network components of the association system.

## Differentiation is associated with preserved cognition in the cognitively healthy elderly

We found that association system segregation and modularity had positive, significant relationships with overall cognition while mean connectivity and participation coefficient did not have a significant relationship. Participation coefficient captures some degree of segregation by reflecting how a node connects to different communities. However, it focuses on the average behavior of individual nodes rather than treating the network as a whole unit. In contrast, segregation and modularity metrics assess the overall network structure and community organization, which may contribute to their greater statistical robustness. Additionally, modularity and segregation were tightly related. Overall cognition is not correlated with mean connectivity, which shows that measures like segregation and modularity go beyond measurement of network strength and provide insight into how the system is organized and functioning.

When we analyzed specific networks within the association system (FPN, CON, and DMN), we found that the network segregation of the FPN and DMN were related to overall cognition. All networks' segregations were correlated to processing speed performance and the effect size of correlations between processing speed and FPN and DMN segregation were similar. We have shown that association network segregation is related to overall cognitive abilities and one of the key cognitive functions affected by aging: processing speed. The findings of our study support the dedifferentiation hypothesis since the association system and its networks do not function as well when they are not differentiated adequately.

Prior studies have shown that FPN, CON, and DMN properties relate to processing speed task performance (*Madden et al., 2010*; *Malagurski et al., 2020*; *Reineberg et al., 2015*; *Rieck et al., 2021b*). Recent research indicates that the FPN regulates other brain networks to support cognitive functioning (*Avelar-Pereira et al., 2017*; *Marstaller et al., 2015*). The FPN and DMN interact less efficiently in older adults compared to younger adults; the networks are coupled during rest and across tasks in older adults, suggesting that aging causes the FPN to have more difficulty flexibly engaging and disengaging networks (*Avelar-Pereira et al., 2017*; *Grady et al., 2016*; *Spreng and Schacter, 2012*). Age-related within-network structural changes and between-network functional dedifferentiation may disrupt the FPN's ability to control other networks, like the DMN and CON (*Avelar-Pereira et al., 2017*; *Geerligs et al., 2015*; *Grady et al., 2016*; *Marstaller et al., 2015*; *Romero-Garcia et al., 2014*; *Zhang et al., 2014*). Because of the FPN's function as a control network, age-related disruptions in FPN connectivity may explain the initial and most noticeable difference in cognition, processing speed (*Ng et al., 2016*; *Oschmann and Gawryluk, 2020*; *Rieck et al., 2021a*). Our results suggest that processing speed might be linked to the maintained segregation of key brain networks, specifically the DMN, the CON, and the FPN. This implies that despite the overall trend toward dedifferentiation that occurs in aging, effective cognitive functioning in the oldest-old could be associated with the continued distinctiveness and specialization of these critical networks. These findings highlight the importance of network organization in sustaining cognitive health as we age. It appears that maintaining a certain level of network segregation—where different brain networks retain their separable functional connectivity—could be a key factor in supporting healthy cognitive aging. This underscores the potential role of specific network organization patterns in preserving cognitive abilities, even in the presence of broader age-related changes in brain function. However,

while our results provide valuable insights, they also point to the need for further research. To fully understand how network segregation is maintained and its impact on cognitive health, future studies should investigate the underlying mechanisms that support the stability of network organization in healthy aging. This includes exploring how various factors might contribute to the preservation of network segregation.

While segregation is not the only metric that can detect differentiation, our findings indicate that it reliably relates to cognitive abilities. With segregation's connection to cognition, it may serve as a more sensitive metric than other network metrics when assessing cognition in aging populations. Additionally, our work helps inform other research that has indicated that segregation may be a marker of potential cognitive resilience in Alzheimer's disease (*Ewers et al., 2021*) and prior work has begun to investigate its usage as a marker for future cognitive status (*Chan et al., 2021*). Studies have shown that learning-induced plasticity through cognitive training and exercise could be an avenue for changing network dynamics to improve cognitive performance (*Iordan et al., 2017*; *Voss et al., 2010*). Future research could target network dynamics in the older adult population to preserve cognitive functioning.

## Limitations and future directions

Since this work is based on data collected across multiple sites, the data collection site was used as a covariate in partial correlation analysis. Across analyses, inclusion of site as a covariate had little to no effect on statistical tests. However, we recognize that this may not completely address site differences, such as different test administrators, different populations, and scanner inhomogeneities. While there are many potential confounds to fMRI in older adults samples (age-related vascular changes, volume loss, changes to white matter integrity, etc.), in this work we included cortical thickness values as a covariate to account for the potential confounding of fMRI signal due to atrophy in this oldest-old sample. We performed post-collection data quality assessment methods, including visual inspection of MRI and cognitive data, strict fMRI preprocessing steps, visual inspection of all generated surfaces and motion parameters, and double data entry for all cognitive data.

We also recognize that the generalizability of our findings is limited due to the limited diversity of our sample which is mostly non-Hispanic, Caucasian, and highly educated individuals. Prior work has shown that these factors can influence association system segregation (*Chan et al., 2021*). Given the cross-sectional nature of this work, we have limited information about our participants' state of health and cognitive performance earlier in life or what their cognitive health will be later in life. Thus, we are not able to investigate whether an individual's current cognitive performance differs from prior performance or if they will go on to develop cognitive impairment.

Future work could expand on this study by (1) broadening the diversity of oldest-old samples, (2) investigating longitudinal changes in cognition and functional networks to evaluate differences in rates of decline among the oldest-old, (3) investigating the interplay between the association networks and how their segregation from each other and possibly other specific networks is associated with cognitive performance, (4) investigating the mechanisms of change in functional network segregation in aging, and (5) investigate the role of variability in structural metrics such as white matter integrity and cortical area as potential moderators of the observed relationships.

We would also like to make clear that the scope of this work is focused on healthy oldest-old age and is cross-sectional in nature. Therefore, inferences from this study focus on what we can learn from individuals who survived to 85+ and are cognitively healthy in their oldest-old years. We have discussed the benefits of studying this age group above.

## Conclusions

This work provides novel insight into the healthy oldest-old brain and intact cognition in aged individuals. We add to the literature on age-related dedifferentiation, showing that (1) in a very old and cognitively healthy sample, differentiation is related to cognition. This suggests that previously observed relationships are not due to inclusion of participants with early stage disease. Further, (2) the segregation of individual networks within the association system is related to a key cognitive domain in aging: processing speed. These findings have theoretical implications for aging. Better cognitive aging seems to be related to a narrow range of relatively high neural network segregation. This effect is specific to the relationship of processing speed to elements of the association networks. These

**Table 4.** Descriptive statistics of characteristics of the study sample.

| Participant characteristics | Total sample, *N* = 146 |
| --- | --- |
| Age (years), mean ± SD (range) | 88.4 ± 3.18 (85–99) |
| Education (years), mean ± SD (range) | 16.1 ± 3.03 (9–26) |
| *Sex, N(%)* | |
| Female | 79 (54.11%) |
| Male | 67 (45.89%) |
| *Race, N (%)* | |
| Non-Hispanic Caucasian | 134 (91.78%) |
| African American | 6 (4.11%) |
| Hispanic Caucasian | 5 (3.42%) |
| Asian | 1 (0.69%) |
| *Marital status, N (%)* | |
| Widowed | 74 (50.69%) |
| Married | 54 (36.99%) |
| Divorced | 13 (8.90%) |
| Living as married/domestic partnership | 3 (2.06%) |
| Never married | 2 (1.37%) |
| *Dominant hand, N (%)* | |
| Right | 131 (89.73%) |
| Left | 15 (10.27%) |

findings inform the broader conceptual perspective of how human brain aging that is normative vs. that which is pathological might be distinguished.

## Materials and methods
### Participants

Data were collected as part of the MBAR, funded by the Evelyn F. McKnight Brain Foundation. Data were collected from the four McKnight Institutes: the University of Alabama at Birmingham, the University of Florida, the University of Miami, and the University of Arizona. The study sample includes 197 individuals with cognitive data and 146 with cognitive and MRI data, after excluding 10 participants due to high head movement in MRI, 6 due to anatomical incompatibility with Freesurfer surface rendering, and 1 due to outlier network segregation values. Participants were community-dwelling, cognitively unimpaired older adults, 85–99 years of age. We performed a multi-step screening process, including exclusions for memory disorders, neurological disorders, and psychiatric disorders. Details of the screening process are shown in *Figure 7—figure supplement 1*. In the first stage of screening, trained study coordinators administered the Telephone Interview for Cognitive Status modified (TICS-M) (*Cook et al., 2009*) and conducted an interview to determine whether the patient met major exclusion criteria, which included individuals under age 85, presence of MRI contraindications, severe psychiatric conditions, neurological conditions, and cognitive impairment. The telephone screening was followed by an in-person screening visit at which eligible participants were evaluated by a neurologist, a comprehensive medical history was obtained to ascertain health status and eligibility, and the Montreal Cognitive Assessment (MoCA) was administered (*Nasreddine et al., 2005*). Participants were recruited through mailings, flyers, physician referrals, and community-based recruitment. Participant characteristics are shown in *Table 4*. Participant characteristics of the full sample of 197 participants used in the cognitive data analysis can be found in *Supplementary file 1* broken down by data collection site. Informed consent was obtained from all participants and approval for the study was

received from the Institutional Review Boards at each of the data collection sites including University of Alabama at Birmingham (IRB protocol X160113004), University of Florida (IRB protocol 201300162), University of Miami (IRB protocol 20151783), and University of Arizona (IRB protocol 1601318818).

## Cognitive measures

Multiple imputation is a statistical technique to estimate missing values in a dataset by pooling multiple iterations of possible values for missing data (*Murray, 2018*; *Nassiri et al., 2018*). While missingness in our dataset was minimal, we chose to still impute data in order to avoid other more simplistic avenues to address missing data such as listwise deletion or replacing values with the mean. Missing values included Stroop interference score (10 missing values), Trails B score (3 missing values), and Stroop word trial score (6 missing values). Missingness was due to administrator error, participant's inability to correctly perceive the stimuli due to low visual acuity or color blindness, or the participant not finishing the Trails B task in the allotted time. We acknowledge that these sources of missing data are not considered missing-at-random; however, they are also not uncommon in neuropsychological data collection and are at a relatively low level of missingness. We obtained a similar mean and range of the variables when the dataset was restricted to only complete cases. An exploratory factor analysis with varimax rotation was performed on 18 variables to identify cognitive domains. The exploratory factor analysis used all available cognitive data (n = 196). The number of factors was determined by eigenvalue greater than 1, analysis of scree plot, and parallel analysis, which indicated five factors (*Humphreys and Montanelli Jr., 1975*; *O'connor, 2000*; *Zwick and Velicer, 1986*). Factor scores were then calculated using the regression method (*Thomson, 1939*). Cognitive measures used for this exploratory factor analysis can be found in *Supplementary file 2*. Overall cognition was calculated as the average of the factor scores for each individual.

Quality control was performed on behavioral data through REDCap double data entry, wherein data are entered twice, and discrepancies are identified and corrected (*Harris et al., 2019*; *Harris et al., 2009*). Data were also visually inspected for errors.

## Network analysis

### Imaging acquisition

For all subjects, an anatomical scan was collected (T1-weighted; repetition time [TR] = 2530 ms; echo time [TE] = 3.37 ms; field of view [FOV (ap,fh,rl)]=240 × 256 × 176 mm; slice gap = 0; voxel size = 1.0 × 1.0 × 1.0 mm; flip angle [FA] = 7°). After the anatomical scan, an 8-minute resting-state functional scan was collected (T2*-weighted, TE/TR 30/2400 ms; FOV = 140 × 5 × 140; FA = 70°; voxel size = 3.0 × 3.0 × 3.0 mm; interleaved order of acquisition). Before the functional scan, participants were instructed to try to be as still as possible, stay awake, keep their eyes open, and let their minds wander without thinking of anything in particular. A central fixation cross was presented during the scan, which participants were told they could choose to look at during the scan.

### Preprocessing

Anatomical images were preprocessed through Freesurfer (version 6.0) to render cortical surfaces (*Fischl, 2012*). Generated surfaces were then visually inspected for errors.

Before functional connectivity analysis, data were preprocessed with rigorous quality control methods for motion censoring (*Carp, 2013*; *Gratton et al., 2020*; *Power et al., 2012*; *Power et al., 2015*; *Siegel et al., 2014*), implemented by XCPEngine (*Ciric et al., 2018*) and fMRIPrep (*Esteban et al., 2019*). Nuisance regressors included global signal, cerebral spinal fluid, white matter (WM), the six motion parameters, their temporal derivatives, and their first-order and quadratic expansion. Censoring included a framewise displacement threshold of 0.5 mm, a DVARS (derivative of the root mean square) threshold of 5, a high-pass filter of 0.01, and a low-pass filter of 0.08. Spatial smoothing of 6 mm full-width-half-max was applied.

## Network nodes

We build upon *Chan et al., 2014* and *Han et al., 2018* by creating nodes from our oldest-old sample. Since our sample of oldest-old adults was larger and included more fMRI data per participant than *Han et al., 2018* or *Chan et al., 2014*, we generated nodes from our sample using the same methods. *Han et al., 2018* showed that while functional connectivity boundary-based parcellation of the human cortex

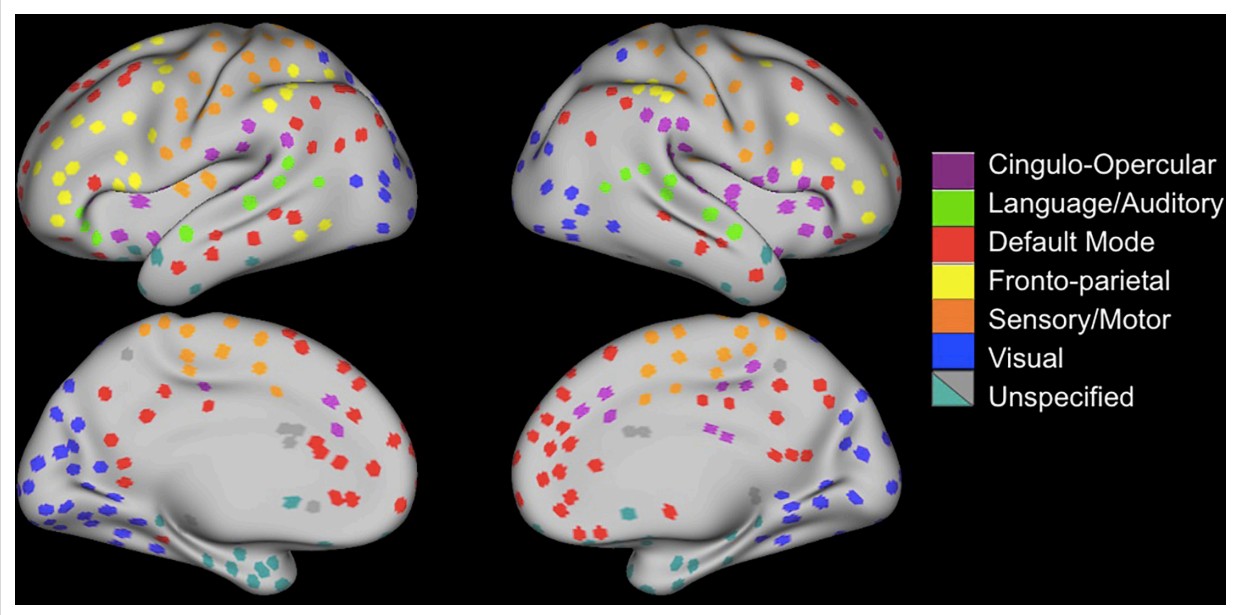

**Figure 7.** Nodes created from community detection of MBAR nodes (*Figure 1*). Sensory/motor system consisted of visual, language/auditory, and sensory/motor networks. Association system consisted of cingulo-opercular, default mode, and fronto-parietal networks.

The online version of this article includes the following figure supplement(s) for figure 7:

**Figure supplement 1.** Participant screening process.

was generally consistent across the lifespan, the boundaries become less similar to the younger adult boundaries as cohorts get older. However, the relationship between increasing age and decreasing system segregation was still intact even with older adult nodes (*Han et al., 2018*). This difference between young and oldest-old adult parcellations led us to use the same methods of boundary-based parcellation as *Han et al., 2018* (*Figure 1A*), the method of detection of local minima ROIs and creation of 3 mm radius discs as *Chan et al., 2014* (*Figure 1B*), and network membership identification from the parcellation by *Power et al., 2011* (*Figure 1C*) to assess system and network segregation.

In order to assess the replicability of our findings, we additionally did the same analyses of the relationship between processing speed and segregation using alternative parcellations including those created by *Chan et al., 2014* and *Han et al., 2018*. We also created node network assignments using community detection with the Louvain algorithm, implemented by the Brain Connectivity Toolbox (*Rubinov and Sporns, 2010*) with gamma level set to 1.2 (*Figure 7*).

## Calculation of network properties

In each participant, a mean time course was computed for each node from the atlas. A node-to-node correlation matrix was formed by correlating each node's time course with every node (*Figure 2*). The matrix of Pearson's r values was then transformed into Fisher's z. Only positive correlations were retained for all metrics except the within-network mean connectivity for which both negative and positive values were incorporated. Within-network connectivity was calculated as the mean node-to-node z-value of all the nodes within that network. Segregation was calculated as within-network connectivity minus between-network connectivity, divided by within-network connectivity (*Chan et al., 2014*; *Wig, 2017*). Association system segregation refers to the average segregation of networks within that system and network segregation refers to the segregation of that network from other networks within the same system (e.g., segregation of the FPN would be the segregation of the FPN from other association system networks). Participation coefficient and modularity were calculated using the Brain Connectivity Toolbox (*Rubinov and Sporns, 2010*).

## Covariates

Cortical thickness was used as a covariate because in elderly populations, there is more likelihood of age-related brain changes such as atrophy. Since we are measuring fMRI signals in the gray matter,

atrophy could influence the strength of those signals. Therefore, including cortical thickness, the thickness of the gray matter, as a covariate, is essential for accounting for possible individual differences in gray matter due to atrophy. Cortical thickness data was derived from Freesurfer's cortical surfaces. Cortical thickness values were averaged across all relevant nodes for the system/network of interest in each analysis. This variable was then used as a covariate in analyses in order to account for potential confounding effects of atrophy. Additionally, *Figure 1—figure supplement 1* shows the relationship between age and cortical thickness which was not significant.

Since data were collected across multiple sites, site-related differences in data collection could occur. Though we took substantial measures to mitigate this potential bias (testing administration training and quality control, MRI sequence homogenization, and frequent assessments of drift throughout data collection), we included site of data collection as a covariate in analyses.

## Relating cognition to network metrics

We performed correlation analysis between overall cognition and each association system metric, as well as cognitive domains (processing speed, executive functioning, episodic memory, working memory, and language) and network segregation (FPN, DMN, and CON). A negative control of the sensory/motor system for the relationship between processing speed and association system and networks was included. Partial correlations with the site and cortical thickness as a covariate were assessed, and FDR correction was used for multiple comparison correction. Multiple linear regressions were performed as additional supplemental analyses (*Supplementary file 5*).

## Acknowledgements

Thank you to all those that helped with data collection and data management from the MBAR collaborative team. Thank you to all of the participants for volunteering their time and energy in contributing to this study, without whom this type of research would be impossible. Thank you to UAB Research Computing and other members of the Visscher lab. Thank you for funding by the Evelyn F McKnight Brain Foundation. SAN was funded through the NIH/NINDS T32NS061788-12 07/2008.

## Additional information

### Funding

| Funder | Grant reference number | Author |
| --- | --- | --- |
| Evelyn F. McKnight Brain Research Foundation | | Ron A Cohen<br>Bonnie E Levin<br>Tatjana Rundek<br>Gene E Alexander<br>Kristina M Visscher |
| National Institute of Neurological Disorders and Stroke | T32NS061788-12 07/2008 | Sara A Nolin |

The funders had no role in study design, data collection and interpretation, or the decision to submit the work for publication.

### Author contributions

Sara A Nolin, Conceptualization, Data curation, Formal analysis, Funding acquisition, Investigation, Visualization, Methodology, Writing – original draft, Project administration; Mary E Faulkner, Formal analysis, Funding acquisition, Visualization, Writing – original draft, Writing – review and editing; Paul Stewart, Formal analysis, Supervision, Funding acquisition, Writing – review and editing; Leland L Fleming, Data curation, Formal analysis, Visualization, Writing – original draft; Stacy Merritt, Formal analysis, Project administration; Roxanne F Rezaei, Mary Kate Franchetti, Project administration; Pradyumna K Bharadwaj, Emily J Van Etten, Theodore P Trouard, Adam J Woods, Data curation; David A Raichlen, Cortney J Jessup, Data curation, Project administration; Lloyd Edwards, Data curation, Formal analysis, Writing – review and editing; G Alex Hishaw, Conceptualization, Data curation,

Funding acquisition; David Geldmacher, Data curation, Writing – review and editing; Virginia G Wadley, Conceptualization, Funding acquisition, Writing – review and editing; Noam Alperin, Bonnie E Levin, Conceptualization, Data curation, Funding acquisition, Writing – review and editing; Eric S Porges, Data curation, Funding acquisition; Ron A Cohen, Data curation, Funding acquisition, Writing – review and editing; Tatjana Rundek, Funding acquisition, Project administration; Gene E Alexander, Formal analysis, Funding acquisition, Writing – review and editing; Kristina M Visscher, Conceptualization, Data curation, Supervision, Funding acquisition, Project administration, Writing – review and editing

### Author ORCIDs
Leland L Fleming ![ORCID] https://orcid.org/0000-0002-4047-9031
Kristina M Visscher ![ORCID] https://orcid.org/0000-0003-0737-4024

### Ethics
Informed consent was obtained from all participants and approval for the study was received from the Institutional Review Boards at each of the data collection sites including University of Alabama at Birmingham (IRB protocol X160113004), University of Florida (IRB protocol 201300162), University of Miami (IRB protocol 20151783), and University of Arizona (IRB protocol 1601318818).

### Decision letter and Author response
Decision letter https://doi.org/10.7554/eLife.78076.sa1
Author response https://doi.org/10.7554/eLife.78076.sa2

## Additional files

### Supplementary files
Supplementary file 1. Participant characteristics.

Supplementary file 2. Factor loadings for cognitive domains.

Supplementary file 3. Correlations between processing speed and segregation in each parcellation.

Supplementary file 4. Correlations between cognitive domains and network segregation.

Supplementary file 5. Supplemental regressions.

Transparent reporting form

### Data availability
Code is available for node creation at https://github.com/Visscher-Lab/MBAR_oldestold_nodes (copy archived at *Visscher-Lab, 2025a*) and code and post processed data for statistical analyses and figures is available at https://github.com/Visscher-Lab/MBAR_segregation_paper (copy archived at *Visscher-Lab, 2025b*). Because these data come from a select group of people who have lived to oldest-old ages, making them potentially identifiable, raw data is not available. More detailed data than the post processed data available online can be requested by submitting a request with explanation of intended use of the data to kmv@uab.edu. Requests are reviewed by a committee of principal investigators of the McKnight brain aging registry.

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
