## [Editor Report]

This useful study provides solid support for how brain function at the system level, particularly network segregation, influences cognitive abilities even in the oldest-old range of human aging. The findings are potentially interesting to help understand successful aging.

---

## [Decision Letter]

**Decision letter after peer review:**

Thank you for submitting your article "Fronto-Parietal Network Segregation Predicts Maintained Processing Speed in the Cognitively Healthy Oldest-old" for consideration by *eLife*. Your article has been reviewed by 2 peer reviewers, and the evaluation has been overseen by a Reviewing Editor and Timothy Behrens as the Senior Editor. The following individual involved in review of your submission has agreed to reveal their identity: Joshua Goh (Reviewer #1).

Essential Revisions (for the authors):

1. Methodology:

a. Cohort-specific parcellation: although it might be more specific to the age group and the study, given the sample size of 146, it is also noisy and less reliable compared to those derived from a large cohort of high-resolution data. Suggest repeating the analyses using a predefined functional parcellation and compare with the current results. This will also allow some comparisons with other age groups (see below).

b. Perform additional control analyses (including other networks and structural measures) to support the claim on the specific network involved in oldest-old

c. Hierarchical regression and partial correlation

d. Site differences and missing data

2. Conceptual design and interpretation:

a. Fronto-parietal network versus the default mode network in terms of correlations with processing speed (Figure 5): need to justify the conclusion of the fronto-parietal network only

b. Dedifferentiation versus compensatory: need to include task-fMRI data, which might be hard. Suggest include another age group (middle-aged or youngest-old) for comparison. Substantial revision of the discussion to tune down the argument on dedifferentiation (as the data does not directly support that) and focus on individual differences in cognition, expand network specialization, and control for structural differences.

c. Explain why the oldest old is unique (and) and what new theoretical insights this study provides on top of the existing literature on aging.

*Reviewer #1 (Recommendations for the authors):*

The authors should elaborate more in section 2.1 about the multiple regressions used and the variables involved for the power calculation. It is noted that the report in section 2.1 of the assessment of power of 0.8 to detect significant associations is not clear what multiple regression variables were included. Section 4.2 which is referenced here does not seem to lend much detail towards understanding the specifics of the power analysis conducted.

Lines 237 – 244. The statement that the best predictor of processing speed among the networks is FPN might need to be qualified or better justified. In Figure 5, the network segregation with the highest correlation with processing speed is the DMN (0.273) compared to the FPN (0.272). Thus, the authors need to quality the above statement.

There are various language and grammatical errors throughout that the authors should take further care to fix.

Expressions in the text that suggest that the findings represent maintenance or changes in cognitive and brain aging might be avoided in this cross-sectional, limited age range sample data. That being said, while the value of this study is in the examination of oldest-old and the limitation noted in section 3.4, still the conclusions and interpretability of the findings are limited by a lack of any comparison made, either in terms of additional data or by way of other similar studies, with data on young, middle-aged or younger-old adults. The authors need to address this major limitation in a more compelling manner, given their motivation to look at this age group in the first place. For instance, in the conclusion, it is highlighted that dedifferentiation is found even in oldest-old healthy adults. However, the authors did not present any suggested reason why one might suspect that dedifferentiation would be different in this age group; they should do so.

The authors might consider direct comparisons of the effects of the network measures on cognitions rather than the current approach which examines each network contribution more or less separately. If the differences in the contributions or effects of each network relative to the others is not significantly different, it is somewhat misleading to highlight that a given network was the "best" predictor. It is more appropriate to highlight the significant contributions of the other networks in more equal ranking. Indeed, looking at the scatterplots, one comes away with the general effect of network segregation on cognition, with minimal differences between networks.

In addition, the inclusion of the Hand, Mouth, or Sensory Motor systems in the paper seems redundant and should be addressed. The authors should consider evaluating cognition in relation to these other networks and also include some control analyses. Currently, the findings are all positivistic (i.e., geared towards supporting a positive effect without a negative control).

The justification for the use of cortical thickness as a covariate from which to compare the effects of the other variables in hierarchical regression needs to be more comprehensively justified.

The issue of site differences in brain functional signals should also be dealt with in order to address possible effects driven by site differences (or the compromising effect on detecting significant results).

*Reviewer #2 (Recommendations for the authors):*

1) It is certainly careful of the authors to include cortical thickness as a covariate. I think it would be useful to include a summary of global/regional cortical thickness distribution as a function of age, or even regional analyses (which could be used as more spatially restrained covariate than the wholebrain average), e.g., as supplementary materials.

2) It would be great if the authors can elaborate a bit more on the imputation procedure for non-expert like myself (despite similar conclusions with and without imputed data. Some of the conditions (e.g., failure due to visual acuity and time limit) do not sound 'missing at random' to me.

3) Often, age is also included as a covariate (Chan, 2014; Ng, 2016) to claim a general effect within the cohort. Did the authors also consider such models?

4) I may be too picky, but I couldn't tell how the 'forward selection' was performed. From Methods it sounds like all fMRI metrics were added to the models simultaneously as a 2nd block in the hierarchical regressions? Is this addition of fMRI metrics to the covariate model the forward selection step? To me the word 'forward selection' is not very informative with such a simple model. I was expecting some selection was done among the various fMRI metrics.

4) Network integration is used in an ambiguous way in Introduction (p3). The definition on p3 sounds more like network specialization to me (which the authors also used, appropriately, throughout the manuscript), since network 'integration' often refers to the (increased) coupling between networks; and it seemed to be readily replaced by 'specialisation' in the next paragraph.

5) minor typos (e.g. p8 line 216 'remained'?) here and there

[Editors' note: further revisions were suggested prior to acceptance, as described below.]

Thank you for resubmitting your work entitled "Network Segregation Predicts Processing Speed in the Cognitively Healthy Oldest-old" for further consideration by *eLife*. Your revised article has been evaluated by Timothy Behrens (Senior Editor) and a Reviewing Editor.

The manuscript has been improved but there are some remaining issues that need to be addressed, as outlined below:

1. Clarify the conceptual framework about oldest-old and discuss the oldest-old findings with reference to literature and dedifferentiation hypothesis

2. Rewrite the Results section and focus on the major findings (reorganization is needed)

3. Pay attention to the writing and revision format to improve readability

*Reviewer #2 (Recommendations for the authors):*

The authors have applied a major revision to this manuscript. Key in this revision is the focus on network segregation as an index of age-related neural dedifferentiation. Much of the introductory and results text has been replaced.

I do appreciate the authors' extensive work on this revision. However, there are still many critical concerns regarding this manuscript in its present state. One big problem is that the revised manuscript is still quite disorganized and very hard to read so it is difficult to distill what is the critical conceptual knowledge gap that their data fills. The authors now focus more on the notion of dedifferentiation in oldest-old. They argue that this provides better support that individual differences in dedifferentiation are present in the oldest-old, and greater differentiation is related to better processing speed. Yet, I had asked before, is there any conceptual basis for us to think that this association would not be the case? Moreover, past studies have already evaluated this, albeit not in the oldest-old sample. Thus, it really is difficult to be convinced of the conceptual novelty of this study. The authors have not addressed this concern. I suggest the authors need to provide a better argument about what potential theoretical alternatives there are regarding the oldest-old sample with respect to the association between brain network segregation and generic processing speed. Also, what reason might we have for considering that oldest-old brains are more or less segregated than younger-old brains? Finally, what reason might we have for considering that segregation degree in oldest-old might be positively or negatively associated with processing speed?

Also, although the authors state that they are focusing on network segregation and processing speed, their results still present various other graph theoretic indices, and various cognitive different cognitive domains, and consider segregation between and within various different brain networks. The authors need to present in the introduction the theoretical argument of the expected difference between the between vs. within network results, the different cognitive domains, and the different graph theoretic indices. Otherwise, this begs the question of why there is a need to consider these various aspects in this study. Moreover, it seems that the authors are suggesting that processing speed mediates the effects of brain network segregation on the various cognitive domains. If so, the authors need to consider adopting more sophisticated statistical models, such as using structural equation models.

Finally, it would be greatly appreciated if the authors could submit a clearer revised manuscript format. The current format includes deleted words, with the formatting across pages very messy. For a review to be more efficient, I suggest that the detailed revision notes and deleted words should be hidden, the figures properly adjusted to fit the relevant paragraph or page, and all paragraphs and headers should be indented correctly. Essentially, the authors should take care of their overall manuscript format to aid readability. The figure and table captions are rather brief as well. Overall, the writing style tends to start each sentence with a present participle (i.e., '*ing'). Moreover, each sentence comes across as an independent or flanking conceptual thread that lacks a smooth connection from the previous sentence. These are style issues that are not immediately problematic to the central paper, but I think at the moment, the language used does compromise the quality of the paper still.

Some further specific comments follow.

1. Line 25. The first "healthy" can be removed.

2. Line 28. "Cortical association system". I am not aware of this label or system. Please provide a suitable nomenclature system that defines this "cortical association system". Are you referring to different association systems in the brain? This is different from the whole CON, FPN, DMN considered as one broad association system.

3. Line 37. "Experiencing" might be better replaced with "representing".

4. Lines 57-58. "… because of their advanced age and the normal age and related plasticity processes.…" reads very awkwardly.

5. Lines 123-134. The mechanistic linkage between age-related dedifferentiation, reduction of segregation of neural network modules, and cognitive processing needs to be better conceptualized in this paragraph (and in the introduction). The authors have reworked their argument to focus on the network segregation index. However, with this more specific focus, a more mechanistic view is needed to avoid this study being an evaluation of associations based on affordance rather than a justified theory. This paragraph attempts to do that, but it merely lists the brain regions and cognitive domains implicated in previous studies. I suggest that the notion of dedifferentiation is more specific than is depicted in this paragraph. For instance, the authors need to elaborate on how and why dedifferentiation might be associated with FPN, CON, and DMN segregation. Critically, how and why is the segregation of neural network modules theorized to be mechanistically linked to these cognitive domains via processing speed? Functional dedifferentiation of neural networks occurs at different scales with different mechanisms. It is likely an oversimplification to consider all these the same phenomenon across these different studies and operating in the same in different networks.

6. Section 2.1. This statement of power needs to be elaborated on. How was it determined that the sample size can detect small effect sizes at the given parameters? What is the justification that the effect sizes expected would be small? I think the authors are trying to provide this information, but these sentences need to be expressed more clearly.

7. Figures. All acronyms used in the figures need to be specified in the captions.

8. Tables. The formatting is not very clear and the table captions or headers need to provide more details to help the reader understand the variables, the numbers, and the reason motivating the presentation of this data or information.

9. Table 1, Lines 219 to 229, Figure 3. If the hypothesis was about segregation, why was there a need to evaluate differences between graph-theoretic metrics? The other measures were not presented in the Introduction and should not be included in the results if not of focus. At present, their inclusion here is rather confusing.

10. Throughout the results, the authors present the statistical results without correction for multiple comparisons, then state which results survive upon correction. I suggest that if the results do not survive multiple comparisons, then the authors might simplify matters and not regard them at all. Only report results that survive multiple comparisons. Alternatively, the authors could provide clearer hypotheses for specific analyses in which they expected certain effects, which might possibly bypass the need for multiple comparison adjustments.

11. Throughout the results as well, there are several sections that are essentially single sentences. I think this should be avoided. The authors need to rework their manuscript writing to better fit the format if they wish to present Methods at the end rather than prior to the Results section. Also, if the Methods are placed at the end, then more basic technical details might be moved to the Results to aid readers to grasp the contexts of the analyses conducted and the statistical findings. Presently, just with reading the results, it is difficult to know what was the research aim of conducting the analyses reported and what hypothesis was addressed with the resulting findings.

*Reviewer #3 (Recommendations for the authors):*

The authors have responded quite adequately to most of the comments. They have now focused on the extension of the positive association between cognitive abilities and brain network segregation, in particular the high-level associative networks, to the oldest olds, who are believed to be a case of truly healthy aging, alluding to the dedifferentiation hypothesis. Supplemented analyses using several related parcellation methods reiterated the importance of network heterogeneity / variations on understanding neurocognitive relationships.

Despite the improvements, I still feel that the manuscript a bit underwhelming.

1) The unique values of the oldest old is multifold as the authors now presented in greater details; but which is the best frame the readers should use to interpret these findings? Are we regarding the oldest old as the template of healthy ageing (general case)? Or are we trying to understand 'is cognition maintained till very old through the same principle (differentiation) as earlier in the lifespan' (specific case)?

For instance, some of the characteristics of the oldest old alluded to by the authors are debatable. For one, we know (e.g., from the Nun study) cognition can be intact with the presence of really bad conditions (e.g., heavy load of amyloid and tau). If a potential issue of younger cohorts in past studies is that we cannot adequately exclude diseased individuals simply based on cognitive criteria, a similar issue of the oldest old might be that we do not know what keeps them intact (neural / cognitive reserve? neural maintenance? Resilience against pathologies?) simply based on cognitive criteria either. When should we understand the findings as generalizable (enrich the dedifferentiation hypothesis that explains the 'development of aging process') or not (p20, line 499 onwards)? These possibilities are obviously not mutually exclusive but I feel that the authors can discuss/separate them more systematically.

Related, the authors discussed general neurocognitive aging quite thoroughly, but since this is about oldest old, what do we know about oldest old so far? How rich/depleted the current literature is regarding this? The novelty of the current findings should stand out much more when such contrast is present.

2) Part of it is also about the content organization. If the segregation and cognition relationship is the main findings, I would expect it to be discussed first before elaborating on other things (e.g., parcellation)

If the dedicated parcellation is also an important message, I would expect more elaborated discussion (than current section 3.1) with a stronger focus on what it means beyond its availability and replicability when compared to previous parcellations. Same goes to the abstract.

I also don't expect to see 'the goal of the study' 1.5 pages deep into Discussion…

[Editors' note: further revisions were suggested prior to acceptance, as described below.]

Thank you for resubmitting your work entitled "Network segregation is associated with processing speed in the cognitively healthy oldest-old" for further consideration by *eLife*. Your revised article has been evaluated by Timothy Behrens (Senior Editor) and a Reviewing Editor.

The manuscript has been improved but there are some remaining issues that need to be addressed, as outlined below:

*Reviewer 1:*

Thank you for your efforts to respond to the previous comments. I note that additional clarifications have been made regarding the novel contribution, the conceptualization about dedifferentiation, the definitions and motivations behind the analyses and metrics used. Despite these revisions, however, I do still find that my concerns remain.

The argument that looking at oldest-old offers us a way of seeing how healthy aging looks like in brain and cognitive metrics is noted. However, there is not a comparison sample made to non-healthy aging or non-oldest-old healthy aging here, which is critical if that was the motivating research question. The correlations reported in this study between association network segregation and a processing speed factor in oldest-old are thus not compelling nor novel as other studies in non-oldest-old have also shown this. The assumption or alluding to possible incipient disease in non-oldest-old samples, unless explicitly examined, remains just an assumption. The studies on these samples certainly make their case regarding their findings in spite of possible incipient diseases. As such, the issue of novelty and motivation in this present examination of oldest-old, sans explicit comparisons with other samples, still remains.

The description of the different network metrics has been slightly expanded. However, it is still not precisely clear how these metrics, which understandably capture different aspects of brain functional organization, relate to the issue of processing speed, or to the application to examine the oldest-old. Again, the findings on broad correlations are not novel as compared to non-oldest-old, which other studies have already reported and to greater specificity in terms of neural and cognitive mechanisms.

The definition of dedifferentiation is now noted to be complex and varied. However, as I read in the introduction, the description of what dedifferentiation is still seems somewhat inaccurate or at least imprecisely used. Because of a vague definition of dedifferentiation, it is therefore not clear how to view the analyses and results. For instance, perhaps the reduced functional segregation observed might be due to an increased need for neural network computations to cross communicate rather than due to a biological reduction in inhibitory modulation. These are two very different things that need to be dissociated. Without this, it seems to be that the functional segregation index and its association with processing speed remains a not very useful description. However, it is not clear if this study can do this.

Thus, overall, there is a lot of technical expertise displayed in the manuscript with brain parcellation methods and graph theoretic metric derivations. However, the relating of what these numerical or algorithmic metrics mean in the brain, and how they are associated with psychological constructs is still very questionable without sufficient validation and postulating of sufficiently specific mechanisms.

*Reviewer 2:*

Thanks for the additional work by the authors to revise their manuscript. While I remain convinced that this is an important population for deciphering 'successful aging', I share reviewer 1's comments regarding novelty when the key underlying mechanisms or concepts remain mostly assumptions without more direct support from the study. Not just aging studies, but lifespan studies have together established the importance of network segregation. Empirically showing that segregation remains important at extreme age, prevailing most age-related diseases and maximizing individual variations, is informative but the impact is limited without further elucidating the dedifferentiation mechanisms or unravelling extreme-age-specific processes (albeit not the primary interest here).

Some more specific remarks:

1) The purported increase in individual variation is interesting. Why is 'more variability = better capturing of brain-cognition relationship' in FC at this age? Are there meaningful subgrouping or individual differences contributing to this? Why is it not also in structure? e.g. How about the cortical area (genetically divergent from thickness, among other differences) and white matter 'integrity' (e.g., Freesurfer white matter hypo intensity estimates; I expect considerable white matter alterations by this age?). The set of 'confounds' (or potential modulators of FC) at this extreme age might be more extensive than typical younger-old cohorts.

2) I understand the technical contribution of the super ager parcellation, and the authors emphasized the result consistency across parcellations and community definitions, but does this consistency actually tell us anything about aging beyond methodological reliability (e.g., invariance in network organization)? Are there notable differences (or lack thereof) in Chan/Han/Power communities and the super age communities? I can't find any further discussion on this beyond figure 6 and 7.

3) Relatively minor point. Some analyses remain redundant to me. For the correlational analyses, I think the multiple regression including all covariates would suffice. A) The two key covariates are consensus confounds to most functional analyses, I don't see how unadjusted 'raw' FC-cognition correlations enrich our understanding. B) multiple regression is essentially partial correlations; C) I don't see a particular reason to adjust for site and atrophy as separate analyses.

4) If I am not wrong, the participation coefficient also captures some degree of segregation. Are there any thoughts on why it is statistically less robust as segregation/modularity?

Overall, despite the potential value of the study, there are hurdles to overcome.

---

## [Author Response]

Essential Revisions (for the authors):1. Methodology:a. Cohort-specific parcellation: although it might be more specific to the age group and the study, given the sample size of 146, it is also noisy and less reliable compared to those derived from a large cohort of high-resolution data. Suggest repeating the analyses using a predefined functional parcellation and compare with the current results. This will also allow some comparisons with other age groups (see below).

We have repeated the analyses with 3 additional parcellations, or node sets. The initial analysis was performed with a node set created with MBAR data used to define the nodes and Power (2011) atlas was used to determine node network membership. Three alternative node sets were then used in addition to the original analysis, as replication for this revision, in line with your suggestion: (1) Younger adults’ data used to define nodes (Chan 2014) and Power (2011) atlas used to determine node network membership, (2) Older adults data from a different study (Han 2018) used to define nodes and Power (2011) atlas used to determine node network membership, and (3) MBAR data (the current dataset) used to define nodes and MBAR data based community detection used to determine node network membership. These results show that the effects observed are generally replicable while also showing that node sets that are created within the age-appropriate cohort are preferred to younger, age-inappropriate cohort based nodes. This data is shown in Figure 6.

b. Perform additional control analyses (including other networks and structural measures) to support the claim on the specific network involved in oldest-old

We have added a control analysis by using the sensory-motor system which does not overlap with or include the association system or any of our networks of interest. We would also not expect a cognitive process like processing speed to be strongly related to the sensory-motor system, therefore it was a clear choice for a control analysis. This has been added to the text along with Figure 6. There were no statistically significant correlations between the sensory-motor system and processing speed in any node set and effect sizes were minimal, consistent with our original interpretations.

2) We have also added more specificity to the structural measures that we are using as covariates in analyses. Instead of brain-wide cortical thickness, we have included only relevant regions for the system/network of interest. For example, for a partial correlation of the FPN and processing speed, we included the cortical thickness specifically for FPN nodes. We believe this additional specificity adds to the robustness of the cortical thickness covariate.

3)Additionally we have considered the reviewer’s comments and decided to alter the focus of the text from FPN specifically and to instead discuss the association networks together in relationships with cognition. This has been reflected in the title, abstract, and the discussion.

c. Hierarchical regression and partial correlation

We have simplified our statistical approach including focusing on partial correlations, and added more detail to the text to clarify the confusion on these statistical approaches.

d. Site differences and missing data

The reviewers are correct that site differences are an important variable to control. We used the site of data collection as a covariate in partial correlations. Site had no impact on any results.The section on missing data includes reasons for missingness, which we do not claim to be missing-at-random, however the level of missingness is minimal with therefore minimal impact of the potential bias and is not unusual for neuropsychological data. We have used imputation to address missing data. We have further clarified this in the text.

2. Conceptual design and interpretation:a. Fronto-parietal network versus the default mode network in terms of correlations with processing speed (Figure 5): need to justify the conclusion of the fronto-parietal network only

We appreciate the reviewers’ careful thought about the interpretation and conceptual design of the paper. We have done a major rewrite of the paper in order to take into account the conceptual reframing that the reviewers’ comments suggest. We also went through all analyses again carefully and identified a typo in the code which mis-labeled the CON segregation metrics (networks are listed as numbers, and the number for CON was one-off in one script), all analyses were rerun to ensure accurate reporting. All code is available on our github link listed in the manuscript. Minimal changes to results were necessitated by this re-analysis, with the exception of a significantly stronger relationship between CON segregation and processing speed. This has impacted our interpretation of findings regarding network segregation and processing speed. Based on the stronger relationship for the CON network this showed, as well as the reviewer suggestions, we instead focus on the segregation of the association networks more generally, instead of singling out the FPN. We now discuss the association networks together, and their relationships to cognition. This is reflected in the title and discussion. We agree that this reframing makes the interpretation much clearer.

b. Dedifferentiation versus compensatory: need to include task-fMRI data, which might be hard. Suggest include another age group (middle-aged or youngest-old) for comparison. Substantial revision of the discussion to tune down the argument on dedifferentiation (as the data does not directly support that) and focus on individual differences in cognition, expand network specialization, and control for structural differences.

In the original submission, we noted relevant literature which describes both the dedifferentiation hypothesis and the compensation hypothesis of aging. Our original aim was to include more of a literature review of cognitive aging theories in the introduction and discussion, but that choice made it too confusing (and honestly left out much important literature). In responding to the reviews we realized that bypassing this cursory literature review here is preferable for the readability of the manuscript. Instead, we cite a literature review, and focus on the dedifferentiation hypothesis.

The data we show here addresses the dedifferentiation hypothesis specifically since we are using the segregation metric which is a reflection of dedifferentiation of network organization. The reviewers’ comments caused us to do a great deal of thinking on this topic, and we have a forthcoming review with our colleague Ian McDonough that covers this topic in more detail (McDonough, Nolin, Visscher, 2022). We have substantially rewritten the relevant sections in the discussion (especially section 3.2) to be more clear for readers.

In order to address the suggestion of comparison with other age groups, we have now described prior reported findings of Association System segregation from other lifespan studies in the manuscript as well as in Figure 6- supplemental figure 1, which directly compares metrics in our study to published values. This addition helps to provide context for segregation across the lifespan and how our oldest-old sample fits into other aging research in this area.

Given the confusion regarding terminology the reviewers bring up, we have made edits for clarity regarding the terms. The inclusion of the word “specialization”, albeit briefly, was misleading and confusing for reviewers. We have completely taken this word out of the text since our work is not related to prior work on specialization in the sense of brain region selectivity of responses to stimuli or tasks. We hope this clarifies that we are instead focused on network organization as in prior work such as Chan et al., 2014. In addition, as stated above in response to 1B, we have used cortical thickness as a covariate in analyses to account for structural differences (described in the text in section 4.3.5). Further, at the suggestion of reviewer 2, we have added to that section additional information about these covariates.

c. Explain why the oldest old is unique (and) and what new theoretical insights this study provides on top of the existing literature on aging.

We have added information regarding the utility of studying the oldest-old to the Discussion section. This passage is as follows “Our study of this oldest-old sample fills in gaps of prior aging research. (1) Prior studies have excluded an ever-growing portion of the older adult population when studying network dynamics (these studies typically focus on people under age 85). We have extended prior methods in network dynamics to the oldest-old age range to better understand how aspects of cognition are related to brain networks in the context of healthy aging. (2) In studies of young-older adults (65-80), undetectable pre-symptomatic disease can confound results. In our sample of the healthy oldest-old, we can be confident in their status as successful agers since they are cognitively unimpaired at a late age. (3) More variability in cognitive and brain network variables makes it easier to observe across-subject relationships (Christensen et al., 1994; Gratton et al., 2022). (4) The healthy oldest-old represent the acme of cognitive aging since they have managed to reach expected lifespan without typical levels of diminished cognitive health or developed cognitive disorders. This work gives us insight into the brain functioning of these relatively rare individuals and helps guide our understanding of how cognition is preserved into late ages.”

We have also added to the discussion regarding new theoretical insights our study provides that go beyond existing literature. For example, the Conclusions section now reads, “This work provides novel insight into the healthy oldest-old brain and intact cognition in aged individuals. We add to the literature on age-related dedifferentiation, showing that (1) in a very old and cognitively healthy sample, dedifferentiation is related to cognition. This suggests that previously observed relationships are not due to inclusion of participants with early stage disease. Further, (2) the segregation of individual networks within the association system is related to a key cognitive domain in aging: processing speed. These findings have theoretical implications for aging. First, better cognitive aging seems to result from a narrow range of relatively high neural network segregation. This effect is specific to the relationship of processing speed to elements of the association networks. These findings inform the broader conceptual perspective of how human brain aging that is normative vs. that which is pathological might be distinguished.”

Reviewer #1 (Recommendations for the authors):The authors should elaborate more in section 2.1 about the multiple regressions used and the variables involved for the power calculation. It is noted that the report in section 2.1 of the assessment of power of 0.8 to detect significant associations is not clear what multiple regression variables were included. Section 4.2 which is referenced here does not seem to lend much detail towards understanding the specifics of the power analysis conducted.

The section 2.1 on power analysis has been changed since the hierarchical regressions were removed from the manuscript. Therefore this section only contains information relevant to the correlations. “Using the sample size of 146, all analyses can detect small effect sizes with an α of.05 and a power of.80. The smallest detectable effect for a correlation was r=.23, similar to the effect size found by Chan et al. (2014).”

Lines 237 – 244. The statement that the best predictor of processing speed among the networks is FPN might need to be qualified or better justified. In Figure 5, the network segregation with the highest correlation with processing speed is the DMN (0.273) compared to the FPN (0.272). Thus, the authors need to quality the above statement.

Similarly to essential reviews point #2, we have considered the reviewer’s comments and decided to alter the focus of the text from FPN specifically and discuss the association networks together in relationships with cognition. This has been reflected in the title and the discussion.

There are various language and grammatical errors throughout that the authors should take further care to fix.

We have gone through the manuscript with careful eye to grammatical errors for this revision.

Expressions in the text that suggest that the findings represent maintenance or changes in cognitive and brain aging might be avoided in this cross-sectional, limited age range sample data. That being said, while the value of this study is in the examination of oldest-old and the limitation noted in section 3.4, still the conclusions and interpretability of the findings are limited by a lack of any comparison made, either in terms of additional data or by way of other similar studies, with data on young, middle-aged or younger-old adults. The authors need to address this major limitation in a more compelling manner, given their motivation to look at this age group in the first place. For instance, in the conclusion, it is highlighted that dedifferentiation is found even in oldest-old healthy adults. However, the authors did not present any suggested reason why one might suspect that dedifferentiation would be different in this age group; they should do so.

We appreciate the reviewer’s point, and in response to the suggestion to compare to other age groups, we compared our results to prior reported findings of Association System segregation in other age groups. This is discussed more in our response to essential revision 2b. There is now a description in the manuscript as well as a supplemental figure (Figure 6- supplemental figure 1) relating our findings to prior reported findings of Association System segregation from other lifespan studies. This addition helps to provide context for segregation across the lifespan and how our oldest-old sample fits into other aging research in this area.

We now more fully describe the reasoning behind studying healthy agers in the discussion. This section is the second paragraph of the discussion and is reproduced above in “essential revisions part 2.c.” This point is also addressed in the first paragraph of the introduction.

Regarding the section 3.4 that was referenced by the reviewer, the last paragraph of that section now reads “We would also like to make clear that the scope of this work is focused on healthy oldest-old age and not the developmental process of aging. Therefore, inferences from this study focus on what we can learn from individuals who survived to 85+ and are cognitively healthy in their oldest-old years. We have discussed the benefits of studying this age group [in the second paragraph of the discussion] above.” Analyses comparing age groups are outside the scope of this dataset and our research questions. Here we simply make the point clear to readers that this is not a study that compares age groups.

Regarding the reviewer’s point about the original paper highlighting that segregation is found in this age group, we originally mentioned that we can measure segregation is found in the oldest-old age group because it has not been previously studied. Our inclusion of the word “even” gave an impression we did not mean to convey. We have removed the sentence from the conclusion, as we think the point we meant to make (segregation can be measured in oldest old adults, and there is a range of segregation values in that population) is made sufficiently elsewhere in the sections referenced in the previous paragraphs.

The authors might consider direct comparisons of the effects of the network measures on cognitions rather than the current approach which examines each network contribution more or less separately. If the differences in the contributions or effects of each network relative to the others is not significantly different, it is somewhat misleading to highlight that a given network was the "best" predictor. It is more appropriate to highlight the significant contributions of the other networks in more equal ranking. Indeed, looking at the scatterplots, one comes away with the general effect of network segregation on cognition, with minimal differences between networks.

Our response to this point is similar to essential reviews point #2; in response to the reviewer’s comments, we altered the focus of the text away from focusing on the FPN specifically and discuss the association networks together in relationships with cognition. This has been reflected in the title and the discussion.

In addition, the inclusion of the Hand, Mouth, or Sensory Motor systems in the paper seems redundant and should be addressed. The authors should consider evaluating cognition in relation to these other networks and also include some control analyses. Currently, the findings are all positivistic (i.e., geared towards supporting a positive effect without a negative control).

We have added a control analysis using the sensory-motor system which does not overlap with or include the association system or any of our networks of interest. We would also not expect a cognitive process like processing speed to be strongly related to the sensory-motor system, therefore it was a clear choice for a control analysis. This has been added to the text along with Figure 6. There were no statistically significant correlations between the sensory-motor system and processing speed in any node set and effect sizes were minimal. Thank you for the suggestion, we agree that the addition of this control bolsters the argument.

The justification for the use of cortical thickness as a covariate from which to compare the effects of the other variables in hierarchical regression needs to be more comprehensively justified.

We have added this further explanation to the text in section 4.3.5: “Cortical thickness was used as a covariate because in elderly populations, there is more likelihood of age-related brain changes such as atrophy. Since we are measuring fMRI signals in the grey matter, atrophy could influence the strength of those signals. Therefore, including cortical thickness, the thickness of the grey matter, as a covariate, is essential for accounting for possible individual differences in grey matter due to atrophy.”

The issue of site differences in brain functional signals should also be dealt with in order to address possible effects driven by site differences (or the compromising effect on detecting significant results).

The reviewer is correct that site differences are an important variable to control. We used the site of data collection as a covariate in partial correlations. In addition, site was added in step 1 for hierarchical regressions (these hierarchical regressions were taken out of the current version of the paper, and are only reported in the supplemental materials (Supplementary file 5), in order to streamline our analyses based on responses to reviewer 2). Site had no impact on any results. This has been stated more explicitly in section 4.3.5: “Since data were collected across multiple sites, site-related differences in data collection could occur. Though we took substantial measures to mitigate this potential bias (testing administration training and quality control, MRI sequence homogenization, and frequent assessments of drift throughout data collection), we included site of data collection as a covariate in analyses.” In addition, we explicitly stated that site was a covariate in correlation analyses in the methods, section 4.3.6.

Reviewer #2 (Recommendations for the authors):1) It is certainly careful of the authors to include cortical thickness as a covariate. I think it would be useful to include a summary of global/regional cortical thickness distribution as a function of age, or even regional analyses (which could be used as more spatially restrained covariate than the wholebrain average), e.g., as supplementary materials.

Thanks to the reviewer for getting us thinking along these lines. We included a summary of cortical thickness distribution as a function of age. Based on the direction of thinking that this reviewer comment brought us to, we have further specified the cortical thickness regions to be specific to the system or network in the analysis. Therefore, instead of the whole brain average, only the thickness of the DMN would be a covariate in a DMN/processing speed correlation for example. We hope this specificity helps improve the functionality of cortical thickness as a covariate. We have also included an age by cortical thickness distribution in supplemental materials (Figure 1-Supplemental Figure 1).

2) It would be great if the authors can elaborate a bit more on the imputation procedure for non-expert like myself (despite similar conclusions with and without imputed data. Some of the conditions (e.g., failure due to visual acuity and time limit) do not sound 'missing at random' to me.

The section on missing data includes reasons for missingness, which we do not claim to be missing-at-random, however the level of missingness is minimal with therefore minimal impact of the potential bias and is not unusual for neuropsychological data. We have used imputation to address missing data.

Briefly, imputation is a method used to resolve missing data in datasets. It is commonly used and multiple imputation is superior to other simpler methods such as replacement with the mean or list-wise deletion (removing the participant from the dataset). Multiple imputation instead generates possible data to “fill in” the missing data points through a series of regression models. These iterations are then pooled and the complete dataset can be used for statistical analysis. We have further explained this in the text to clarify.

3) Often, age is also included as a covariate (Chan, 2014; Ng, 2016) to claim a general effect within the cohort. Did the authors also consider such models?

We didn’t include age as a covariate because the age range in our study was very limited since this study was intentionally focused on the oldest-old age range (85-99 years of age). Therefore, lack of variability in the covariate of age would likely cause it to not be an impactful variable for our analyses. We opted to instead focus on other likely influencing variables such as site of data collection and atrophy.

4) I may be too picky, but I couldn't tell how the 'forward selection' was performed. From Methods it sounds like all fMRI metrics were added to the models simultaneously as a 2nd block in the hierarchical regressions? Is this addition of fMRI metrics to the covariate model the forward selection step? To me the word 'forward selection' is not very informative with such a simple model. I was expecting some selection was done among the various fMRI metrics.

Based on the reviewers’ feedback about making the conceptual aspect of the paper cleaner and stronger, we decided to remove the forward selection model from the analysis.

5) Network integration is used in an ambiguous way in Introduction (p3). The definition on p3 sounds more like network specialization to me (which the authors also used, appropriately, throughout the manuscript), since network 'integration' often refers to the (increased) coupling between networks; and it seemed to be readily replaced by 'specialisation' in the next paragraph.

Thank you for noting that potentially confusing use of the term. After thinking about your comment, we decided to use the term “within-network integration” instead. We think this term is much cleaner. In the following paragraph in the original manuscript, the term ‘specialization’ muddied the waters. We removed that word, and refer to (and define) segregation alone. We now state in the Introduction “Within-network integration describes how much the network’s regions interact and can be quantified as the mean connectivity of nodes within a given network (within-network connectivity).”

[Editors’ note: what follows is the authors’ response to the second round of review.]

The manuscript has been improved but there are some remaining issues that need to be addressed, as outlined below:1. Clarify the conceptual framework about oldest-old and discuss the oldest-old findings with reference to literature and dedifferentiation hypothesis

We provide additional clarification about the purpose and novelty of the study and why the oldest-old sample is used and how it helps answer our research questions regarding cognitive aging and dedifferentiation.

2. Rewrite the Results section and focus on the major findings (reorganization is needed)

We have made extensive edits to the Results section to add clarity (linking the hypotheses and predictions to the results and linking specific methods sections to their specific results).

3. Pay attention to the writing and revision format to improve readability

We have made extensive edits to improve clarity in the writing. We agree that a “clean” version of the manuscript, without mark-up for our many modifications, is easier to read. For this reason, we will submit both the clean version and the marked-up version that is required to identify changes.

Reviewer #2 (Recommendations for the authors):The authors have applied a major revision to this manuscript. Key in this revision is the focus on network segregation as an index of age-related neural dedifferentiation. Much of the introductory and results text has been replaced.I do appreciate the authors' extensive work on this revision. However, there are still many critical concerns regarding this manuscript in its present state. One big problem is that the revised manuscript is still quite disorganized and very hard to read so it is difficult to distill what is the critical conceptual knowledge gap that their data fills. The authors now focus more on the notion of dedifferentiation in oldest-old. They argue that this provides better support that individual differences in dedifferentiation are present in the oldest-old, and greater differentiation is related to better processing speed. Yet, I had asked before, is there any conceptual basis for us to think that this association would not be the case? Moreover, past studies have already evaluated this, albeit not in the oldest-old sample. Thus, it really is difficult to be convinced of the conceptual novelty of this study. The authors have not addressed this concern. I suggest the authors need to provide a better argument about what potential theoretical alternatives there are regarding the oldest-old sample with respect to the association between brain network segregation and generic processing speed. Also, what reason might we have for considering that oldest-old brains are more or less segregated than younger-old brains? Finally, what reason might we have for considering that segregation degree in oldest-old might be positively or negatively associated with processing speed?

We have added to the introduction to more clearly state the goals of the studies and explain its novelty and purpose. We hope this addition makes it more clear earlier on in the manuscript what we are achieving in this work.

Additionally, we have provided background literature supporting that (a) oldest-old brains are less segregated than younger-old brains in the introduction (see paragraph 4 in the introduction) and (b) network organization properties such as segregation are associated with cognitive abilities (see paragraph 5 in the introduction). While prior work has been usually limited in its representation of the 85+ age range, we believe our work uniquely helps to expand our knowledge of aging to include the full spectrum of the human life course.

Also, although the authors state that they are focusing on network segregation and processing speed, their results still present various other graph theoretic indices, and various cognitive different cognitive domains, and consider segregation between and within various different brain networks. The authors need to present in the introduction the theoretical argument of the expected difference between the between vs. within network results, the different cognitive domains, and the different graph theoretic indices. Otherwise, this begs the question of why there is a need to consider these various aspects in this study. Moreover, it seems that the authors are suggesting that processing speed mediates the effects of brain network segregation on the various cognitive domains. If so, the authors need to consider adopting more sophisticated statistical models, such as using structural equation models.

In order to address the comment regarding provided theoretic support and our purpose in including other graph theoretical indices, we have added the following:

In the introduction paragraph 4 it now includes the theoretical reasoning for including the network indices in the analysis to further clarify their purpose. It now states “In our analyses, we incorporate segregation along with other network organization metrics. The aim is to not only examine whether any general network organization metric is associated with cognition in the oldest-old but also to specifically explore if there is evidence supporting dedifferentiation.”A specific prediction statement has been added to the last paragraph of the introduction as it now states “We predict that segregation will be related to cognition and that other network organization metrics will have relatively weaker associations.”

Given our sample size, unfortunately we are not sufficiently powered for analyses such as structural equation modeling. We have specifically avoided misleading statements regarding our statistical approach such as not referring to it as “mediation”. This type of analysis is referred to in the future directions section of the manuscript.

Finally, it would be greatly appreciated if the authors could submit a clearer revised manuscript format. The current format includes deleted words, with the formatting across pages very messy. For a review to be more efficient, I suggest that the detailed revision notes and deleted words should be hidden, the figures properly adjusted to fit the relevant paragraph or page, and all paragraphs and headers should be indented correctly. Essentially, the authors should take care of their overall manuscript format to aid readability. The figure and table captions are rather brief as well. Overall, the writing style tends to start each sentence with a present participle (i.e., '*ing'). Moreover, each sentence comes across as an independent or flanking conceptual thread that lacks a smooth connection from the previous sentence. These are style issues that are not immediately problematic to the central paper, but I think at the moment, the language used does compromise the quality of the paper still.

We have made additions to the captions to enhance clarity and detail for figures. We have also made revisions to reduce use of present participles.

It is our understanding based on communication with the *eLife* office staff that we are to resubmit manuscripts in the given format with mark-up. We agree that a “clean” version of the manuscript would be helpful, however we have complied with the requirements that *eLife* staff have requested of us.

Language of the paper has generally been revised for clarity.

Some further specific comments follow.1. Line 25. The first "healthy" can be removed.

Removed.

2. Line 28. "Cortical association system". I am not aware of this label or system. Please provide a suitable nomenclature system that defines this "cortical association system". Are you referring to different association systems in the brain? This is different from the whole CON, FPN, DMN considered as one broad association system.

Changed “cortical association system” to simply “association system” since that is how it is referred to in the rest of the paper and will add to consistency of the language. As in the rest of the paper, the Association System is a broader system that has sub-networks: FPN, CON, and DMN.

3. Line 37. "Experiencing" might be better replaced with "representing".

This edit has been made.

4. Lines 57-58. "… because of their advanced age and the normal age and related plasticity processes.…" reads very awkwardly.

We have modified this sentence to be clearer, and it now reads, “Studying successful cognitive agers brings another advantage: given the aging process, as well as the years of experience they have due to their advanced age, there is greater variability in both their performance on neurocognitive tasks and their brain connectivity measures compared to younger cohorts. (Christensen et al., 1994). This increased variance makes it easier to observe across-subject relationships of cognition and brain networks (Gratton, Nelson, & Gordon, 2022).

5. Lines 123-134. The mechanistic linkage between age-related dedifferentiation, reduction of segregation of neural network modules, and cognitive processing needs to be better conceptualized in this paragraph (and in the introduction). The authors have reworked their argument to focus on the network segregation index. However, with this more specific focus, a more mechanistic view is needed to avoid this study being an evaluation of associations based on affordance rather than a justified theory. This paragraph attempts to do that, but it merely lists the brain regions and cognitive domains implicated in previous studies. I suggest that the notion of dedifferentiation is more specific than is depicted in this paragraph. For instance, the authors need to elaborate on how and why dedifferentiation might be associated with FPN, CON, and DMN segregation. Critically, how and why is the segregation of neural network modules theorized to be mechanistically linked to these cognitive domains via processing speed? Functional dedifferentiation of neural networks occurs at different scales with different mechanisms. It is likely an oversimplification to consider all these the same phenomenon across these different studies and operating in the same in different networks.

The reviewer makes a good point that the concept of dedifferentiation can be a thorny one. We operationalize it here using the measure of segregation. However, dedifferentiation has been shown in other ways, including dedifferentiation of stimulus driven signals. Segregation is related to cognitive processing and the mechanistic linkage between age-related dedifferentiation and cognitive processing can be difficult to describe. As the beginning of the referenced paragraph states, these networks have been shown to be involved in dedifferentiation and that this has been related to decline in cognitive abilities including processing speed, among others. As stated in paragraph 2 of the introduction, there is reason to believe that processing speed is related to all these other cognitive domains, as stated “Salthouse (1996) proposed that cognitive aging is associated with impairment in processing speed, which in turn may lead to a cascade of age-associated deficits in other cognitive abilities. Because processing speed is so strongly associated with a wide array of cognitive functions, it is crucial to understand how it can be maintained in an aging population”. Therefore, there is justification for analyzing processing speed as an essential cognitive function that would be impacted by the dedifferentiation of cognitive networks.The referenced paragraph (previously lines 128-139) then continues to describe the networks. This description serves as an introduction to the networks of interest and what has been found regarding activation of these networks usually in the context of a task requiring specific cognitive demands. This section is essential for readers who may be unfamiliar with these canonical networks and what role they play in cognition.Regarding the reviewers second paragraph in comment #5, if referring to the studies cited in the first statement, functional dedifferentiation is a theoretical framework that has been tested and studied using many different kinds of methods. We do not claim that all the referenced works are all using the exact same methods or “scales” or “mechanisms”, however they are all supporting the same idea of dedifferentiation occurring in functional networks, which is what the statement is aiming to supply background on. If referring to the task activation studies referenced in the rest of the paragraph, these studies were about activation and not dedifferentiation and their citations here are to provide background of foundational work about what is known on the functionality of these networks.

6. Section 2.1. This statement of power needs to be elaborated on. How was it determined that the sample size can detect small effect sizes at the given parameters? What is the justification that the effect sizes expected would be small? I think the authors are trying to provide this information, but these sentences need to be expressed more clearly.

The section on power analysis has been reworded for clarity.

7. Figures. All acronyms used in the figures need to be specified in the captions.

These have been added, especially in figure 2.

8. Tables. The formatting is not very clear and the table captions or headers need to provide more details to help the reader understand the variables, the numbers, and the reason motivating the presentation of this data or information.

Tables 1-3 are standard descriptive statistics for the variables analyzed (mean, standard deviation, and range) and a table caption has been added. Table 4 consists of descriptions of the sample and a table caption has been added.

9. Table 1, Lines 219 to 229, Figure 3. If the hypothesis was about segregation, why was there a need to evaluate differences between graph-theoretic metrics? The other measures were not presented in the Introduction and should not be included in the results if not of focus. At present, their inclusion here is rather confusing.

See response to comment 2 for edits that include details regarding need for inclusion of additional graph theoretical indices. These graph theoretical indices are described with their relevant citations in the introduction in paragraph 3.

10. Throughout the results, the authors present the statistical results without correction for multiple comparisons, then state which results survive upon correction. I suggest that if the results do not survive multiple comparisons, then the authors might simplify matters and not regard them at all. Only report results that survive multiple comparisons. Alternatively, the authors could provide clearer hypotheses for specific analyses in which they expected certain effects, which might possibly bypass the need for multiple comparison adjustments.

Regarding the reporting of multiple comparisons, we report the results from all the analyses we performed as well as the results from multiple comparison correction in order to be forthcoming and transparent about what exactly we did in the analyses. We have made edits to clarify our hypotheses in the manuscript, however we believe multiple comparison corrections are still needed.

11. Throughout the results as well, there are several sections that are essentially single sentences. I think this should be avoided. The authors need to rework their manuscript writing to better fit the format if they wish to present Methods at the end rather than prior to the Results section. Also, if the Methods are placed at the end, then more basic technical details might be moved to the Results to aid readers to grasp the contexts of the analyses conducted and the statistical findings. Presently, just with reading the results, it is difficult to know what was the research aim of conducting the analyses reported and what hypothesis was addressed with the resulting findings.

Thank you for your comment that will aid in readability of the paper. In order to help with clarity of the relationship between the results and the methods while not increasing redundancy in the work, we have added referencing to specific methods sections within the Results section. For example, “We created network nodes based on methods developed by Chan et al. (2014) and Han et al. (2018) for our oldest-old sample (Figure 1, methods section 4.3.3)”.

In addition, the predictions as stated in the last paragraph of the introduction have been restated in the Results section to aid the reader’s ability to know what part of the hypothesis is being addressed in the relevant Results section.

Reviewer #3 (Recommendations for the authors):The authors have responded quite adequately to most of the comments. They have now focused on the extension of the positive association between cognitive abilities and brain network segregation, in particular the high-level associative networks, to the oldest olds, who are believed to be a case of truly healthy aging, alluding to the dedifferentiation hypothesis. Supplemented analyses using several related parcellation methods reiterated the importance of network heterogeneity / variations on understanding neurocognitive relationships.Despite the improvements, I still feel that the manuscript a bit underwhelming.1) The unique values of the oldest old is multifold as the authors now presented in greater details; but which is the best frame the readers should use to interpret these findings? Are we regarding the oldest old as the template of healthy ageing (general case)? Or are we trying to understand 'is cognition maintained till very old through the same principle (differentiation) as earlier in the lifespan' (specific case)?For instance, some of the characteristics of the oldest old alluded to by the authors are debatable. For one, we know (e.g., from the Nun study) cognition can be intact with the presence of really bad conditions (e.g., heavy load of amyloid and tau). If a potential issue of younger cohorts in past studies is that we cannot adequately exclude diseased individuals simply based on cognitive criteria, a similar issue of the oldest old might be that we do not know what keeps them intact (neural / cognitive reserve? neural maintenance? Resilience against pathologies?) simply based on cognitive criteria either. When should we understand the findings as generalizable (enrich the dedifferentiation hypothesis that explains the 'development of aging process') or not (p20, line 499 onwards)? These possibilities are obviously not mutually exclusive but I feel that the authors can discuss/separate them more systematically.Related, the authors discussed general neurocognitive aging quite thoroughly, but since this is about oldest old, what do we know about oldest old so far? How rich/depleted the current literature is regarding this? The novelty of the current findings should stand out much more when such contrast is present.

Regarding comment 1, paragraph 1: This work provides insight into both the general and specific case. We have pursued the specific case of investigating “the underlying brain network relationships associated with preserved cognition in oldest old adulthood”. And regarding the general case, this work helps the field of aging research better understand what healthy aging can look like and what a healthy aging brain can do.

Regarding comment 1, paragraph 2: The limitation of studying the young adult cohorts is that they all have potential for disease, however we cannot differentiate who will go on to live into healthy oldest-old adulthood and who will not via cognitive criteria or many of the ways in which we would diagnose disease. We cannot be sure exactly what occurred over the course of their lived experience or potentially genetic predisposition that led our healthy oldest-old individuals to stay intact and this mechanistic approach is outside the scope of the current work. However, given our findings we can propose that segregation is a potential candidate as a reserve/resilience/maintenance factor. Additionally, the reference to the “development of the aging process” in the Discussion section serves as a reminder to readers that this work is cross-sectional in nature and is not able to comment on longitudinal changes in variables. This has been reworded to prevent confusion on the purpose of this statement to the following: “We would also like to make clear that the scope of this work is focused on healthy oldest-old age and is cross-sectional in nature.”

Regarding comment 1, paragraph 3: Thank you for this comment. While work in the oldest-old is limited, we have added to the introduction and discussion prior work that helps highlight the novelty of our work in this age group. For example, we have added the following to the introduction: “Prior work studying the healthy oldest-old indicates intact cognition in this age group is impacted by influences such as cognitive reserve (Kawas et al. 2021) and resistance to Alzheimer’s disease related neuropathology (Biswas et al. 2023; Gefen et al. 2015). We extend oldest-old aging research by increasing our understanding of the oldest-old brain and provide novel insight into the relationship between the segregation of networks and cognition by investigating this relationship in an oldest-old cohort of healthy individuals.”

2) Part of it is also about the content organization. If the segregation and cognition relationship is the main findings, I would expect it to be discussed first before elaborating on other things (e.g., parcellation)If the dedicated parcellation is also an important message, I would expect more elaborated discussion (than current section 3.1) with a stronger focus on what it means beyond its availability and replicability when compared to previous parcellations. Same goes to the abstract.I also don't expect to see 'the goal of the study' 1.5 pages deep into Discussion…

The parcellation is shown earlier in the Results section because it gives context for the rest of the results. It is important to show how we are representing the networks on the brain prior to going into results about their connectivity and segregation. The parcellation is also a contribution of this work to the broader brain aging research community because it provides a valuable resource for future studies, as stated in section 3.1.

We have added to the introduction to more clearly state the goals of the studies and explain its novelty and purpose. We hope this addition makes it more clear earlier on what we are achieving in this work.

[Editors’ note: what follows is the authors’ response to the third round of review.]

The manuscript has been improved but there are some remaining issues that need to be addressed, as outlined below:Reviewer 1:Thank you for your efforts to respond to the previous comments. I note that additional clarifications have been made regarding the novel contribution, the conceptualization about dedifferentiation, the definitions and motivations behind the analyses and metrics used. Despite these revisions, however, I do still find that my concerns remain.The argument that looking at oldest-old offers us a way of seeing how healthy aging looks like in brain and cognitive metrics is noted. However, there is not a comparison sample made to non-healthy aging or non-oldest-old healthy aging here, which is critical if that was the motivating research question. The correlations reported in this study between association network segregation and a processing speed factor in oldest-old are thus not compelling nor novel as other studies in non-oldest-old have also shown this. The assumption or alluding to possible incipient disease in non-oldest-old samples, unless explicitly examined, remains just an assumption. The studies on these samples certainly make their case regarding their findings in spite of possible incipient diseases. As such, the issue of novelty and motivation in this present examination of oldest-old, sans explicit comparisons with other samples, still remains.The description of the different network metrics has been slightly expanded. However, it is still not precisely clear how these metrics, which understandably capture different aspects of brain functional organization, relate to the issue of processing speed, or to the application to examine the oldest-old. Again, the findings on broad correlations are not novel as compared to non-oldest-old, which other studies have already reported and to greater specificity in terms of neural and cognitive mechanisms.The definition of dedifferentiation is now noted to be complex and varied. However, as I read in the introduction, the description of what dedifferentiation is still seems somewhat inaccurate or at least imprecisely used. Because of a vague definition of dedifferentiation, it is therefore not clear how to view the analyses and results. For instance, perhaps the reduced functional segregation observed might be due to an increased need for neural network computations to cross communicate rather than due to a biological reduction in inhibitory modulation. These are two very different things that need to be dissociated. Without this, it seems to be that the functional segregation index and its association with processing speed remains a not very useful description. However, it is not clear if this study can do this.Thus, overall, there is a lot of technical expertise displayed in the manuscript with brain parcellation methods and graph theoretic metric derivations. However, the relating of what these numerical or algorithmic metrics mean in the brain, and how they are associated with psychological constructs is still very questionable without sufficient validation and postulating of sufficiently specific mechanisms.

Thank you for your review of this manuscript. It seems the reviewer appreciates the technical strengths of the paper but feels that the novelty could be enhanced by including comparisons with non-oldest-old groups and further expansion of dedifferentiation and its relationship to network metrics is warranted. We appreciate your concerns about the lack of comparison to non-oldest-old groups. However, due to the unique nature of our study cohort, we do not have other samples for comparison. Although similar correlations have been reported in non-oldest-old samples, our research provides a unique perspective by focusing on the oldest-old. These individuals are at highest risk of cognitive decline of any other age group yet individuals in this sample do not display cognitive decline. This study provides a window into processing speed, a key cognitive domain in age-related decline, and what aspects of brain functioning are related to cognitive health in the oldest-old age group. This allows us to provide valuable insights into healthy aging in this specific population, which we believe adds a unique contribution to the field. While it is a worthwhile endeavor for future research and we have therefore added it to the appropriate section of the discussion regarding future work, our study is not equipped to address the mechanisms of action for changes in segregation. We have expanded on the key concept of dedifferentiation in the introduction to clarify our analysis and provide context for our results.

Reviewer 2:Thanks for the additional work by the authors to revise their manuscript. While I remain convinced that this is an important population for deciphering 'successful aging', I share reviewer 1's comments regarding novelty when the key underlying mechanisms or concepts remain mostly assumptions without more direct support from the study. Not just aging studies, but lifespan studies have together established the importance of network segregation. Empirically showing that segregation remains important at extreme age, prevailing most age-related diseases and maximizing individual variations, is informative but the impact is limited without further elucidating the dedifferentiation mechanisms or unravelling extreme-age-specific processes (albeit not the primary interest here).

Thank you for your thoughtful feedback and for recognizing the importance of studying the oldest-old population in the context of successful aging. We appreciate your concerns regarding the novelty of our study, particularly in relation to the underlying mechanisms of dedifferentiation and extreme-age-specific processes. While our primary focus was on demonstrating the continued relevance of network segregation in the oldest-old, our study is novel in that it examines these brain functions in a population that represents the extreme end of the lifespan, with a larger sample size than most studies in this age group. This allows us to explore how network segregation remains important despite the increased risk of age-related diseases and cognitive decline, providing valuable insights into the mechanisms that may contribute to cognitive resilience in the oldest-old. We acknowledge that further exploration of the mechanisms behind these observations would enhance the impact of our findings. However, given the scope and design of our current study, we are limited in our ability to directly investigate these mechanisms. We agree that this is an important direction for future research and have added to the Discussion section of the manuscript to further highlight it as an area of future work.

Some more specific remarks:1) The purported increase in individual variation is interesting. Why is 'more variability = better capturing of brain-cognition relationship' in FC at this age? Are there meaningful subgrouping or individual differences contributing to this? Why is it not also in structure? e.g. How about the cortical area (genetically divergent from thickness, among other differences) and white matter 'integrity' (e.g., Freesurfer white matter hypo intensity estimates; I expect considerable white matter alterations by this age?). The set of 'confounds' (or potential modulators of FC) at this extreme age might be more extensive than typical younger-old cohorts.

Thank you for your comments. The increased variability in functional connectivity (FC) at extreme age may reflect individual differences in brain resilience, though significantly more data is needed to explore possible subgroups. We focused on FC due to its relevance to brain function, but we agree that incorporating structural measures like cortical area and white matter integrity could offer valuable insights, especially given the expected age-related changes and therefore we have added this as a direction for future research. We also acknowledge that potential confounds in the oldest-old may be more extensive than in younger cohorts, and we have added language to acknowledge confounds beyond changes in cortical thickness.

2) I understand the technical contribution of the super ager parcellation, and the authors emphasized the result consistency across parcellations and community definitions, but does this consistency actually tell us anything about aging beyond methodological reliability (e.g., invariance in network organization)? Are there notable differences (or lack thereof) in Chan/Han/Power communities and the super age communities? I can't find any further discussion on this beyond figure 6 and 7.

Thank you for your feedback and for recognizing the technical contribution of the oldest-old parcellation. We appreciate your point about distinguishing methodological reliability from insights into aging. The consistency observed across different parcellations and community definitions was meant to demonstrate the robustness of our findings. While there are some differences between the Chan/Han/Power communities and the oldest-old communities, these differences were subtle (small shifts in ROIs), and our focus was on ensuring that the results were not dependent on a specific parcellation scheme.

3) Relatively minor point. Some analyses remain redundant to me. For the correlational analyses, I think the multiple regression including all covariates would suffice. A) The two key covariates are consensus confounds to most functional analyses, I don't see how unadjusted 'raw' FC-cognition correlations enrich our understanding. B) multiple regression is essentially partial correlations; C) I don't see a particular reason to adjust for site and atrophy as separate analyses.

We included both FC-cognition correlations and multiple regression analyses to provide a comprehensive view and address different aspects of variability since other earlier readers of this manuscript requested this. We acknowledge that multiple regression captures partial correlations and that adjusting for site and atrophy separately was intended to account for independent sources of variability.

4) If I am not wrong, the participation coefficient also captures some degree of segregation. Are there any thoughts on why it is statistically less robust as segregation/modularity?

Thank you for your question. You are correct that the participation coefficient captures some degree of segregation by reflecting how a node connects to different communities. However, it focuses on the average behavior of individual nodes rather than treating the network as a whole unit. In contrast, segregation and modularity metrics assess the overall network structure and community organization, which may contribute to their greater statistical robustness. We have added this explanation to the manuscript.

Overall, despite the potential value of the study, there are hurdles to overcome.